# nnSVG for the scalable identification of spatially variable genes using nearest-neighbor Gaussian processes

Lukas M. Weber [1], Arkajyoti Saha[2], Abhirup Datta [1], Kasper D. Hansen [1] & Stephanie C. Hicks [1] ✉

Feature selection to identify spatially variable genes or other biologically informative genes is a key step during analyses of spatially-resolved transcriptomics data. Here, we propose nnSVG, a scalable approach to identify spatially variable genes based on nearest-neighbor Gaussian processes. Our method (i) identifies genes that vary in expression continuously across the entire tissue or within a priori defined spatial domains, (ii) uses gene-specific estimates of length scale parameters within the Gaussian process models, and (iii) scales linearly with the number of spatial locations. We demonstrate the performance of our method using experimental data from several technological platforms and simulations. A software implementation is available at https://bioconductor.org/packages/nnSVG.

Spatially-resolved transcriptomics (SRT) refers to recently developed technologies that measure gene expression in either the full transcriptome or up to thousands of genes at near- or sub-cellular resolution along with spatial coordinates of the measurements, either based on (i) tagging messenger RNA (mRNA) molecules with spatial barcodes followed by sequencing[1–4] or (ii) fluorescence imaging-based in situ transcriptomics techniques where mRNA molecules are detected along with their spatial coordinates using sequential rounds of fluorescent barcoding[5,6]. These technologies have been used to study the spatial landscape of gene expression in a variety of biological systems, including the brain[1,7,8], cancer[9], and embryonic development[10].

However, these new platforms also bring new computational challenges[11]. One common analysis task is to identify genes that vary in expression across a tissue, defined as spatially variable genes by Svensson et al.[12] (SVGs). These SVGs can then be further investigated individually as potential markers of biological processes, or used as the input for downstream analyses such as spatially-aware unsupervised clustering[8,13,14] or registering the spatial locations of single-cell RNA sequencing (scRNA-seq) data[11,15,16].

To identify SVGs, one approach is to ignore the spatial coordinates and apply methods that rely only on the gene expression, such as feature selection methods used in the analysis of scRNA-seq data, including highly variable genes (HVGs)[17–19] or deviance residuals from

binomial or Poisson models[20]. A second approach is to use both the gene expression and spatial coordinates to identify genes that vary in expression in a continuous manner, either across the entire tissue or within a priori defined spatial domains, for example, using morphology from histology images, representing a subset of the tissue. Here, we refer to this second set of approaches with continuous spatial variation as methods to detect SVGs. Some examples of these methods include (i) standard spatial statistics measures (Moran's I statistic[21], Geary's C statistic[22]) to rank genes by their spatial autocorrelation, (ii) marked point processes (trendsceek[23]), (iii) Gaussian process (GP) regression (SpatialDE[12], SpatialDE2[24]), (iv) generalized linear spatial models with either an overdispersed Poisson or Gaussian distribution (SPARK[25]) or a zero-inflated negative binomial distribution (BOOST-GP[26]), or (v) nonparametric covariance tests (SPARK-X[27]). These methods make tradeoffs, for example, flexibility in fitting gene-specific parameters (iii, iv) versus improved computational efficiency (v). There are also toolboxes that incorporate some of these methods, for example, MERINGUE[28], Giotto[29], within a larger end-to-end analysis framework.

A third approach is to detect changes in the average expression at all spatial coordinates within one spatial domain relative to the average expression at all spatial coordinates in other domains. We refer to these approaches as methods to detect spatial domain marker genes.

[1]Department of Biostatistics, Johns Hopkins Bloomberg School of Public Health, Baltimore, MD, USA. [2]Department of Statistics, University of Washington, Seattle, WA, USA. ✉e-mail: shicks19@jhu.edu

The spatial domains can be defined a priori, for example using morphology from histology images, or alternatively using an unsupervised clustering algorithm. However, the primary difference between methods to identify SVGs and these approaches are the type of variation in expression. In contrast to methods searching for continuous variation across the tissue (SVGs), these methods search for changes in the mean expression across spatial coordinates within one domain compared to other domains. This is in similar spirit to detecting marker genes between discrete cell populations in scRNA-seq data. An example of this approach is SpaGCN[14], which first uses the spatial coordinates to identify domains in an unsupervised manner and then performs domain-guided differential expression analysis[14] with a Wilcoxon rank-sum test to identify mean-level changes between the spatial domains. Here, we are interested in methods to identify SVGs, not spatial domain marker genes. Therefore, we focus on methods to identify SVGs in this work.

A key distinguishing characteristic among recent methods to identify SVGs is computational scalability. In particular, SPARK-X scales linearly with the number of spatial locations[27], while other methods scale cubically (e.g., SpatialDE[12] and SPARK[25]) or quadratically (SpatialDE2[24]). This is relevant as datasets from the latest SRT platforms such as 10x Genomics Visium[2] and Slide-seqV2[4] contain thousands of spatial locations per tissue sample, with future development moving towards even higher resolution. In addition, a limited set of existing methods, such as SPARK-X, offer the ability to search for continuous variation in expression within a priori defined spatial domains, which can be incorporated as known covariates in statistical models fit for each gene[27]. However, SPARK-X uses the same set of kernels and length scale parameters for all genes, which reduces flexibility to identify SVGs from different biological processes with varying spatial ranges in expression within the same tissue sample. In this work, we aimed to address this limitation and develop a computationally scalable approach to identify SVGs that fits a flexible length scale parameter per gene and also allows taking spatial domains into account.

Here, we describe nnSVG, a method to identify SVGs, which is based on statistical advances in computationally scalable parameter estimation in spatial covariance functions in GPs using nearest-neighbor Gaussian process (NNGP) models[30–32]. First, we introduce an overview of the methodological framework and then we compare our method to other methods in several SRT datasets including from the 10x Genomics Visium[2], Spatial Transcriptomics[1], Slide-seqV2[4], and seqFISH[33] platforms. Our method can search for SVGs across an entire tissue or within a priori defined spatial domains. In addition, unlike existing scalable methods, our approach estimates a gene-specific length scale parameter within the spatial covariance function in the GPs, enabling flexibility in the types of SVGs identified. We demonstrate that our method scales linearly with the number of spatial locations, ensuring the method can be applied to datasets with thousands or more spatial locations. Our methodology is implemented in the nnSVG R package within the Bioconductor framework[34] and can be integrated into workflows using established Bioconductor infrastructure for SRT and scRNA-seq data[17,35].

## Results

### Overview of the nnSVG model and methodological framework

The nnSVG framework fits a nearest-neighbor Gaussian process (NNGP) model[30,31] to the preprocessed expression values for each gene:

$$\mathbf{y} \sim N(\mathbf{X}\boldsymbol{\beta}, \widetilde{\Sigma}(\boldsymbol{\theta}, \tau^2)) \qquad (1)$$

Here, $\mathbf{y} = (y_1, ..., y_N)$ represents the normalized and transformed expression values of gene $g$ (subscript $g = 1, ..., G$ omitted for

simplicity) at the set of $N$ spatial locations $\mathbf{s} = (\mathbf{s_1}, ..., \mathbf{s_N})$, which we assume to be in two dimensions, but may in principle be generalized. The $\widetilde{\Sigma}(\boldsymbol{\theta}, \tau^2)$ term represents the NNGP covariance matrix, which offers a scalable (linear-time and storage) approximation to the covariance matrix $\Sigma(\boldsymbol{\theta}, \tau^2) = C(\boldsymbol{\theta}) + \tau^2 \mathbf{I}$ from a full GP model, which scales cubically in the number of spatial locations. The GP covariance matrix $C(\boldsymbol{\theta}) = (C_{ij}(\boldsymbol{\theta}))$ (also referred to as a kernel) captures the spatially correlated variation and is parameterized by a vector of parameters $\boldsymbol{\theta}$. We assume an exponential covariance function, based on the observation that the widely used squared exponential covariance function (e.g., used in SpatialDE[12]) decays too rapidly with distance in the context of SRT data[36]. The exponential covariance function is defined as:

$$C_{ij}(\boldsymbol{\theta}) = \sigma^2 \exp\left(\frac{-||\mathbf{s_i} - \mathbf{s_j}||}{l}\right) \qquad (2)$$

with covariance parameters $\boldsymbol{\theta} = (\sigma^2, l)$, and where $||\mathbf{s_i} - \mathbf{s_j}||$ represents the Euclidean distance between two spatial locations $\mathbf{s_i}$ and $\mathbf{s_j}$. Here, $\sigma^2$ is the spatial component of variance, and $l$ is referred to as the length scale (or bandwidth) parameter, which controls the strength of decay of correlation with distance. The parameter $\tau^2$ (referred to as the nugget) represents the additional nonspatial component of variance.

The design matrix $\mathbf{X}_{[N \times d]}$ can include up to $d - 1$ covariates representing known spatial domains or other information at each spatial location. The default is $\mathbf{X} = \mathbf{1}_{[N \times 1]}$, representing an intercept, with $\boldsymbol{\beta}$ accounting for the mean expression level. We fit a separate model for each gene and obtain maximum likelihood estimates for the parameters $\boldsymbol{\theta} = (\sigma^2, l)$ and $\tau^2$ using the fast optimization algorithms for NNGP models implemented in the BRISC R package[32]. The main parameter of interest is $\sigma^2$, on which we perform a likelihood ratio (LR) test comparing the fitted model against a classical linear model that assumes $\sigma^2 = 0$ and hence does not account for the spatial correlation in expression. Finally, we rank genes by the estimated LR statistic values and calculate multiple-testing adjusted approximate $p$-values for statistical significance. We provide a ranked list of genes, which can be used to select either (i) an arbitrary number of top-ranked genes for further investigation or to use as input for downstream analyses, or (ii) a set of statistically significant SVGs based on $p$-values adjusted for false discoveries. In addition, we calculate an effect size defined as propSV $= \sigma^2/(\sigma^2 + \tau^2)$, which is the proportion of spatial variance ($\sigma^2$) from the total variance ($\sigma^2 + \tau^2$), as previously defined by Svensson et al.[12].

### Key innovations of nnSVG

The key innovations of nnSVG compared to existing approaches are as follows. First, since we use NNGPs to fit the models for each gene, the computational complexity and runtime of nnSVG scale linearly with the number of spatial locations while retaining a large proportion of the underlying information[30,31]. SPARK-X also achieves linear scalability[27], while earlier methods (e.g., SpatialDE[12], SPARK[25]) scale cubically with the number of spatial locations and are thus infeasible to apply to large datasets. Second, we demonstrate that because nnSVG estimates a gene-specific length scale parameter within the models, it enables the identification of SVGs associated with distinct biological processes with varying spatial ranges in expression within the same tissue sample. This cannot be achieved with methods that either assume a fixed length scale parameter or a combination of models with fixed length scale parameters across genes (e.g., SPARK-X[27]) or ignore the spatial information. Finally, nnSVG can identify SVGs within spatial domains by including the spatial domains as covariates within the model, which can also be done with SPARK-X[27] but not other existing methods (e.g., SpatialDE[12], SPARK[25]).

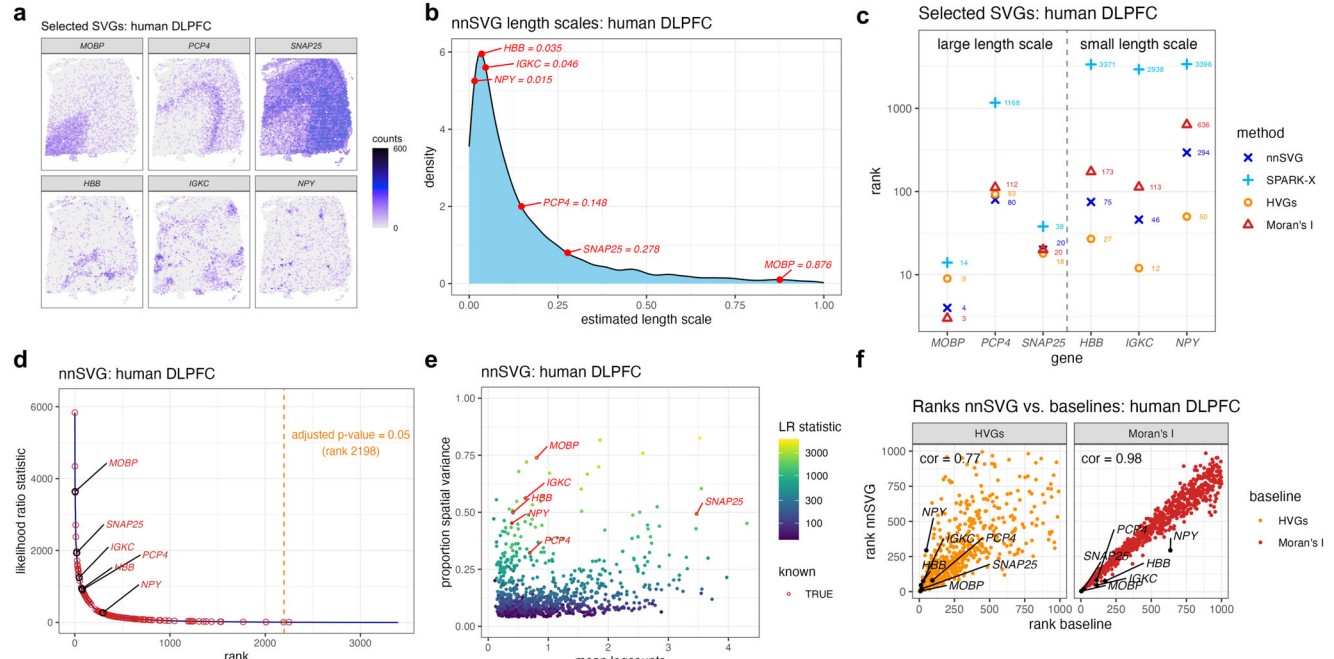

**Fig. 1 | nnSVG recovers biologically informative SVGs with gene-specific length scale parameters.** Using the Visium human DLPFC dataset[8], nnSVG, SPARK-X, HVGs, and Moran's I were applied to identify SVGs. **a** Spatial expression plots of 6 known biologically informative SVGs, including cortical layer-associated SVGs (top row) and blood- and immune-associated SVGs (bottom row). **b** Distribution of estimated gene-specific length scale parameters from nnSVG, with the 6 SVGs from (**a**) labeled in red. The blood- and immune-associated SVGs have smaller estimated length scale parameters than the cortical layer-associated SVGs. **c** Rank order of the 6 SVGs from (**a**) within the lists of top SVGs. Dashed vertical line divides the genes into the 3 cortical layer-associated SVGs with large length scales (left) and the 3 blood- and immune-associated SVGs with small length scales (right). **d** Estimated likelihood ratio (LR) statistic from nnSVG (*y*-axis) compared to the rank per gene (*x*-axis), with the 6 SVGs from (**a**) labeled, and 134 additional known layer-specific marker genes (from manually guided analyses by Maynard et al.[8]) highlighted (red circles). Orange dashed vertical line indicates rank cutoff for statistically significant SVGs at a multiple-testing-adjusted *p*-value of 0.05 using LR test with 2 degrees of freedom. **e** Estimated effect size (proportion of spatial variance) along *y*-axis compared to the mean log-transformed normalized counts (logcounts) along *x*-axis for top 1000 SVGs from nnSVG, with the 6 SVGs from (**a**) labeled, and estimated LR statistic per gene indicated with color scale. **f** Ranks of top 1000 SVGs from nnSVG (*y*-axis) compared to ranks from baseline methods (*x*-axis) using HVGs (nonspatial baseline method, left) and Moran's I (spatially-aware baseline method, right), with SVGs from (**a**) highlighted (black circles), and Spearman correlation (text labels).

## nnSVG recovers biologically informative SVGs with gene-specific length scales

In the following sections, we consider three SRT datasets that contain previously identified biologically informative SVGs: data with (i) variation across cortical layers in the human brain dorsolateral prefrontal cortex (DLPFC)[8] measured with the 10x Genomics Visium platform[37], (ii) variation across cell type layers in the mouse brain olfactory bulb (OB)[1,12] measured with the Spatial Transcriptomics (ST) platform[1], and (iii) variation within a sagittal tissue section of a mouse embryo[38] measured with the seqFISH platform[33]. In the Visium human DLPFC dataset, while the primary variation in expression is across cortical layers, there are also more subtle forms of variation associated with blood vessels and immune processes, which vary in expression across smaller length scales than the main cortical layers[8]. We demonstrate that nnSVG identifies SVGs associated with both forms of variation, and that this flexibility stems from how the nnSVG model fits a gene-specific length scale parameter *l* within the covariance function $C(\theta)$ for each gene (see "Methods"). By contrast, methods that assume a fixed length scale parameter (or a combination of models with fixed length scale parameters) across genes may miss these types of discoveries.

Here, we evaluate the performance of nnSVG in recovering SVGs from the Visium human DLPFC[8], ST mouse OB[1,12], and seqFISH mouse embryo[38] datasets, and compare against SPARK-X[27], which is the only other existing method that also scales linearly with the number of spatial locations, and can therefore be applied to transcriptome-wide datasets with thousands or more spatial locations[27]. In addition, we compare with baseline approaches, specifically HVGs[17] and Moran's I

statistic[21], as nonspatial and spatial baseline methods, respectively (see "Methods").

## nnSVG in application to human brain dorsolateral prefrontal cortex

Based on previously published analyses[8], the Visium human DLPFC dataset is known to contain a number of biologically informative SVGs, including a large number of SVGs associated with cortical layers (Fig. 1a, top row), as well as a smaller set of SVGs associated with blood vessels and immune processes (Fig. 1a, bottom row). The manually labeled cortical layer labels[8] (which we use as an approximate ground truth for method evaluation) are shown in Supplementary Fig. S1A, B as a reference. The spatial expression patterns of the blood- and immune-associated SVGs vary over relatively smaller distance ranges than the cortical layer-associated SVGs, which is reflected by the smaller estimated length scale parameters for the blood and immune-associated SVGs ($\hat{l} < 0.1$) compared to the cortical layer-associated SVGs ($\hat{l} \geq 0.1$) from the nnSVG models (Fig. 1b).

All four methods successfully identified two out of the three cortical layer-associated SVGs (*MOBP* and *SNAP25*) within the top 100 ranked genes. While nnSVG, HVGs, and Moran's I ranked the third SVG (*PCP4*) around rank 100, SPARK-X did not rank *PCP4* within the top 1000 genes (Fig. 1c, left columns). For the three blood and immune-associated SVGs (*HBB*, *IGKC*, *NPY*), we found that HVGs ranked all 3 genes within the top 100, while nnSVG identified two out of the three within the top 100 and the third around rank 300. Moran's I ranked these three genes at lower ranks (ranks ~100–1000), and SPARK-X did

not identify any of the three genes within the top 1000 genes (Fig. 1c, right columns).

To ensure a consistent comparison in these evaluations, we used the same filtering to remove low-expressed genes for both nnSVG and SPARK-X (3396 out of 21,803 genes passed the filtering threshold; see "Methods"). To confirm that the performance of SPARK-X was not affected by the filtering, we also ran nnSVG and SPARK-X without filtering low-expressed genes, in line with the default setting for SPARK-X[27]. The performance for nnSVG was comparable for all six SVGs (with and without filtering). However, the performance for SPARK-X dropped for identifying the blood- and immune-associated SVGs (Supplementary Fig. S2A).

In addition to these 6 SVGs, this dataset also contains a set of 198 known cortical layer-specific marker genes (consisting of 195 additional genes and the 3 cortical layer-associated SVGs from Fig. 1a) identified by manually guided pseudobulked analyses in the original study[8]. Out of these 198 genes, 134 passed filtering for low-expressed genes, and 133 of these 134 were identified as statistically significant SVGs by nnSVG, out of a total of 2198 statistically significant SVGs (from 3396 genes that passed filtering) at an adjusted $p$-value threshold of 0.05. (The likelihood of correctly selecting 133 out of 134 genes by chance in this case is $p < 10^{-16}$ by Fisher's exact test and assuming independently selected genes.) All 3 of the blood- and immune-associated SVGs from Fig. 1a were also included in the set of significant SVGs from nnSVG (Fig. 1d). By contrast, SPARK-X identified 3394 (out of 3396) genes as statistically significant SVGs, including all 134 of the layer-specific markers that passed filtering, but one of the 3 blood- and immune-associated SVGs (*NPY*) was not included within the set of significant SVGs (Supplementary Fig. S2B). Using the default filtering for SPARK-X (i.e., no filtering of low-expressed genes), 10,358 genes were identified as significant SVGs, including 187 out of the 198 layer-specific markers, but not including *NPY* (Supplementary Fig. S2C). Considering the effect size defined as the estimated proportion of spatial variance from nnSVG, following ref. 12 (see "Methods"), we found that highly ranked SVGs (with large LR statistics) also had higher proportion of spatial variance, which is also related to the mean expression (Fig. 1e).

As another form of comparison, we evaluated the degree of overlap between nnSVG and the baseline methods. In this dataset, the main biological signals of interest are related to the spatial distributions of the cortical layers and other biological processes, which are characterized by distinct gene expression profiles[8]. Therefore, we expect that most SVGs will also be identified as HVGs, and thus a strong agreement between nnSVG and HVGs gives further confidence in the results from nnSVG. When comparing the ranks of the top 1000 SVGs from nnSVG and HVGs, we found relatively close agreement (Fig. 1f, left panel), along with a high overlap between the sets of top $n$ SVGs from nnSVG and top $n$ HVGs ($n = 10, 20, 50, 100, 200$) (Supplementary Fig. S2D). We found similar results and higher correlation when comparing between nnSVG and Moran's I (Fig. 1f, right panel), demonstrating that for most genes, these two methods recover similar spatial information. However, the largest mismatch in ranks between nnSVG and Moran's I occurs for the 3 blood- and immune-associated SVGs, especially *NPY*, which have relatively small estimated length scale parameters (Fig. 1b), thus further demonstrating the advantage of the gene-specific length scale parameters in nnSVG and the improved performance in this dataset for genes with small estimated length scales (Fig. 1c). Further investigation of the estimated length scales and effect sizes per gene revealed that nnSVG tends to outperform (i) HVGs for genes with larger length scales ($\geq 0.15$), (ii) Moran's I for genes with smaller length scales ($< 0.15$), and (iii) both baselines for genes with relatively large effect sizes (Supplementary Fig. S3A–C; expression plots of examples of genes where nnSVG outperforms the baselines shown in Supplementary Fig. S3D). In addition, we observe that all genes with extremely small length scales ($< 0.01$)—which may be hard

to estimate reliably—were either not ranked within the top 1000 SVGs or were removed during our filtering step for low-expressed genes ("Methods"), so these genes did not interfere with the final ranking of top SVGs (Supplementary Fig. S4). In contrast, when comparing SPARK-X to the baseline methods, we found smaller overlap and lower correlations using either HVGs and Moran's I (Supplementary Fig. S2E). The SPARK-X results did not substantially change when using default filtering settings for low-expressed genes (Supplementary Fig. S2F).

Finally, we generated spatial expression plots of the top 20 SVGs from nnSVG and SPARK-X, respectively. We observed that most of the top 20 SVGs from nnSVG were related to differences in expression between white matter and gray matter, where gray matter consists of the cortical layers[8] (Supplementary Figs. S1, S5). This is consistent with previous analyses showing that the distinction between white matter and gray matter represents the strongest differences in expression patterns in this dataset, consistent with prior biological knowledge of this brain region[8]. By contrast, the majority of the top 20 SVGs from SPARK-X are not associated as clearly with the distinction between white matter and gray matter (Supplementary Fig. S6).

### nnSVG in application to mouse brain olfactory bulb
The second dataset we considered is the ST mouse OB dataset[1]. This dataset contains a smaller set of spatial locations at lower spatial resolution, which have been annotated with cell type layer labels[1,12] (Supplementary Fig. S7A). Similar to the Visium human DLPFC data, we observe variation in the estimated gene-specific length scale parameters from nnSVG (Supplementary Fig. S7B), suggesting the need for flexibility in this parameter. We considered 7 known layer-associated SVGs[1], and found that HVGs identified all 7 genes within the top 200 ranked SVGs, while nnSVG, Moran's I, and SPARK-X identified 5, 3, and 1 out of the 7 genes within the top 200 ranked SVGs, respectively (Supplementary Fig. S7C). Overall, nnSVG identified 559 genes as statistically significant SVGs (out of 4216 genes that passed filtering) at an adjusted $p$-value threshold of 0.05 (Supplementary Fig. S7D), while SPARK-X identified 2270 (out of 4216) significant SVGs (Supplementary Fig. S7E). When comparing the top 1000 ranked genes between nnSVG and the baseline methods (HVGs and Moran's I), we found a higher correlation (Supplementary Fig. S7F) than when using SPARK-X (Supplementary Fig. S7G). Furthermore, when comparing the overlap between the sets of top $n = 10, 20, 50, 100, 200$ SVGs and HVGs, we found nnSVG had a higher overlap with HVGs compared to the overlap between SPARK-X and HVGs (Supplementary Fig. S7H). This is expected as most of the biologically informative SVGs in this dataset are related to the spatial distribution of cell type layers[1,12]. Finally, we also provide spatial expression plots of the top 20 SVGs from nnSVG (Supplementary Fig. S8) to illustrate that most of the top SVGs are associated with the known cell type layers, as expected.

### nnSVG in application to mouse embryo
The third dataset we considered is the seqFISH mouse embryo dataset[38], which consists of expression measurements of 351 targeted genes summarized at single-cell resolution in a sagittal tissue section from a mouse embryo (Supplementary Fig. S9A). Similar to the other datasets, we observed a range of values for the gene-specific length scale parameters from nnSVG (Supplementary Fig. S9B). We investigated 12 known highly biologically informative SVGs[38], and found that nnSVG gave the highest rankings for 7 of these. In addition, nnSVG, HVGs, and Moran's I identified all 12 of these genes within the top 100 ranked genes, while SPARK-X identified only 9 out of 12 within the top 100 ranked genes (Supplementary Fig. S9C). Overall, both nnSVG and SPARK-X identified all 351 genes as statistically significant SVGs (Supplementary Fig. S9D, E). Similar to the Visium human DLPFC dataset, we also found a relationship between the effect size (proportion of spatial variance) and the mean expression (Supplementary Fig. S9F). When comparing the ranks of the 351 genes between nnSVG or

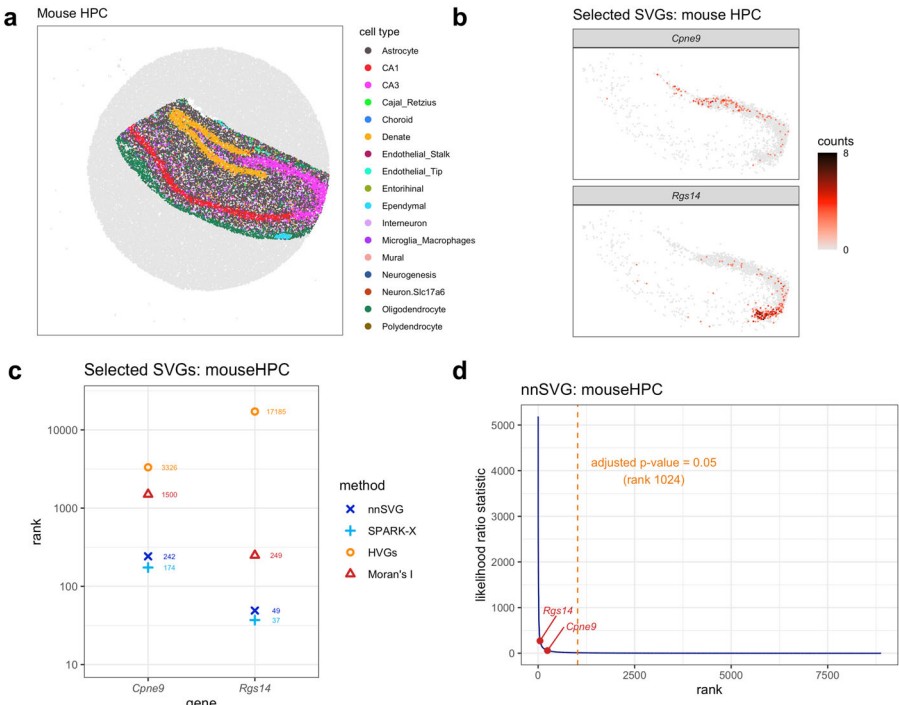

**Fig. 2 | nnSVG recovers biologically informative SVGs within spatial domains.** Using the Slide-seqV2 mouse hippocampus (HPC) dataset[4], nnSVG, SPARK-X, HVGs, and Moran's I were applied to identify SVGs within an a priori defined spatial domain. **a** Computationally labeled cell types per spot (bead) with labels from ref. 39. **b** Spatial expression plots of 2 known biologically informative SVGs identified by Cable et al.[39] showing spatial gradients of expression within the spatial domain defined by CA3 cell type labels (pink points in (**a**)). **c** Rank order of the 2 SVGs from (**b**) within the lists of top SVGs. **d** Estimated likelihood ratio (LR) statistic from nnSVG (*y*-axis) compared to the rank per gene (*x*-axis), with the 2 SVGs from (**b**) highlighted. Orange dashed vertical line indicates rank cutoff for statistically significant SVGs at a multiple-testing-adjusted *p*-value of 0.05 using LR test with 2 degrees of freedom.

SPARK-X and the baseline methods (HVGs and Moran's I), we found a higher correlation for nnSVG than for SPARK-X (Supplementary Fig. S9G, H). Similarly, we found that nnSVG had a higher overlap than SPARK-X between the sets of top $n = 10, 20, 50, 100, 200$ SVGs and HVGs (Supplementary Fig. S9I). As for the other datasets, this is expected since the biologically informative SVGs in this dataset are related to the spatial distribution of cell types at different developmental stages[38]. Finally, we also provide spatial expression plots of the top 20 SVGs from nnSVG (Supplementary Fig. S10).

### nnSVG identifies SVGs within spatial domains

In this section, we apply nnSVG to an SRT dataset to demonstrate how our method can be used to identify SVGs within an a priori defined spatial domain by including the spatial domains as covariates within the statistical models. Specifically, we consider the Slide-seqV2 mouse hippocampus (HPC) dataset[4], which contains separate anatomical regions from the mouse HPC and has been annotated with cell type labels by Cable et al.[39] (Fig. 2a). We highlight two previously identified SVGs (*Cpne9* and *Rgs14*) from ref. 39, which exhibit spatial gradients of expression within the CA3 region of the hippocampus (Fig. 2b). Here, we apply nnSVG, SPARK-X, HVGs, and Moran's I to identify SVGs and compare their performance.

We found that both nnSVG and SPARK-X (which also provides the option to include covariates for spatial domains) rank the *Cpne9* and *Rgs14* genes within the top 300 genes with similar performance (Fig. 2c). In contrast, while Moran's I is able to identify *Rgs14* within the top 300 ranked genes, HVGs does not rank *Rgs14* within the top 1000 ranked genes, and neither HVGs or Moran's I rank the *Cpne9* gene within the top 1000 ranked genes. We note that the performance of the HVGs baseline method differs from the results in the Visium human DLPFC dataset, where HVGs performed well. We also confirmed that using default settings for SPARK-X (no filtering of

low-expressed genes) did not substantially change the results for SPARK-X, although nnSVG performed somewhat worse in this case (genes ranked between top 100−500) (Supplementary Fig. S11A). At an adjusted *p*-value threshold of 0.05, nnSVG identified 1024 genes as statistically significant SVGs (out of 8883 genes that passed filtering), including both *Cpne9* and *Rgs14* (Fig. 2d). SPARK-X identified 2053 (out of 8883) genes as significant SVGs (Supplementary Fig. S11B), or 1809 (out of 21,011) without filtering low-expressed genes (Supplementary Fig. S11C), including both *Cpne9* and *Rgs14* in both cases.

We also compared the performance of nnSVG and SPARK-X to identify SVGs across the entire tissue without incorporating any spatial domain information. This reduced performance for both methods. Both nnSVG and SPARK-X ranked *Cpne9* and *Rgs14* in approximately the top 250 to 1000 genes (compared to the top 300 genes in the models with covariates for the spatial domains) (Supplementary Fig. S11D). In the models without covariates, nnSVG and SPARK-X identified 3217 and 4821 (out of 8883) genes, respectively as statistically significant SVGs (Supplementary Fig. S11E, F). The reduced performance when excluding the spatial domain covariates is expected, since in this case many additional expression patterns are deemed spatially variable.

In order to further evaluate performance for this dataset, we also investigated an extended list of 74 known SVGs within the CA3 spatial domain from the prior analyses[39] (including *Cpne9* and *Rgs14*). We calculated how many of these 74 genes were identified within the top 1000 SVGs or HVGs by each method, which revealed that nnSVG recovered the highest number (27 out of 74), followed by SPARK-X and Moran's I (23 out of 74), while HVGs performed poorly and recovered only 3 out of 74 genes (Supplementary Fig. S12). Finally, we visualized the spatial expression of the top 20 SVGs from both nnSVG and SPARK-X (including spatial domain covariates). For both methods, the

majority of these top SVGs are clearly associated with the known spatial domains, in particular the dentate, CA1, CA3, and oligodendrocyte cell type labels (Supplementary Figs. S13, S14).

## nnSVG to select genes for downstream clustering

One potential application of methods to identify SVGs is to select a list of genes to use as the input for downstream clustering. Using SVGs instead of (nonspatial) HVGs for downstream clustering has been shown to improve clustering performance in SRT datasets[40]. We compared clustering performance in the Visium human DLPFC dataset using either the top 1000 SVGs (nnSVG, SPARK-X, and Moran's I) or the top 1000 HVGs, in terms of the adjusted Rand index (which measures the similarity between two sets of cluster labels, with values between 0 and 1, where 1 indicates perfect agreement) between the clustering and the manually annotated cortical layer labels in this dataset (Supplementary Fig. S15A–D). We used graph-based clustering from standard single-cell workflows[17] and applied the clustering algorithm to the top 50 principal components (PCs) calculated on the top 1000 SVGs or HVGs. The results demonstrated that nnSVG and Moran's I (which take spatial information into account) outperformed HVGs (nonspatial). In addition, nnSVG and Moran's I outperformed SPARK-X, which is consistent with the main results showing that the top SVGs from nnSVG more closely reflect the biological structure in this dataset, compared to SPARK-X (Supplementary Fig. S15E).

## Evaluating nnSVG using simulations

We developed a simulation framework to evaluate the performance of nnSVG in several ways. First, we built a dataset consisting of a set of simulated SVGs with regions of relatively high expression, surrounded by regions of background noise, across several scenarios where we varied the length scale and expression strength. We also simulated a set of noise genes without any spatial expression patterns. We obtained empirical parameters for these scenarios from the Visium human DLPFC dataset—mean and variance of log-transformed normalized expression (logcounts) for the known SVG *MOBP* within the highly expressed region (white matter) and low-expressed region (cortical layers), respectively, as well as proportions of sparsity (zero counts) within both regions. We simulated a total of 1000 genes, consisting of 100 SVGs and 900 noise genes. For the SVGs, we varied the length scale by simulating circular regions of elevated expression with radius 0.25, 0.125, and 0.025 times the width of the tissue section. We varied the expression strength as 1, 1/3, and 1/10 times the average difference between the regions of elevated and low expression, above background noise, for *MOBP*. Supplementary Fig. S16A displays the spatial coordinate masks and relative expression strength for each scenario, and Supplementary Fig. S16B shows the expression values (logcounts). Then, we evaluated the true positive rate (TPR) and false positive rate (FPR) for identifying the simulated subset of SVGs in each scenario. Our evaluations showed that nnSVG achieved very high TPR in all scenarios except the most difficult scenarios with small length scale and medium to low expression strength. In addition, we observed that nnSVG was conservative with respect to false positives—in the medium length scale, medium expression strength scenario (middle panel), we achieved FPR of 0.003, 0.016, and 0.031 at nominal $p$-value thresholds of 0.01, 0.05, and 0.1 (Supplementary Fig. S16C).

We also developed an ablation simulation study[41] to evaluate the robustness of nnSVG to increasing levels of noise. In this simulation, we extended the medium length scale, medium expression strength scenario by randomly shuffling a progressively increasing subset of spatial coordinates (0%, 10%, ..., 100%) to introduce noise into the spatial expression patterns (Supplementary Fig. S17A, B). We evaluated the TPR at each step, and found that nnSVG was highly robust to the noise —the TPR started decreasing at 70% shuffled coordinates, and eventually reached near zero by 90% shuffled coordinates (Supplementary Fig. S17C).

Finally, we performed a set of null simulations using two datasets (Visium human DLPFC and ST mouse OB), where we permuted the order of the spatial coordinates to remove any spatial correlation structure. We observed the spikes in the distributions of the $p$-values near 0 were removed in the null simulations (Supplementary Fig. S18A, B), confirming that the significant $p$-values in the main results are due to the spatial correlation in expression. We also evaluated the proportion of false positives in the null simulations, which further confirmed that nnSVG is relatively conservative and generates a low proportion of false positives. Specifically, we evaluated the error control by calculating the proportion of false positives at a $p$-value cutoff of 0.05, which gave values below the nominal value of 0.05 in both null simulations (1.5% and 1.0%, respectively) (Supplementary Fig. S18C).

## Evaluating the $p$-value distributions from nnSVG

Next, we investigated the $p$-value distributions from nnSVG for the transcriptome-wide datasets presented in this work, prior to correcting for false discoveries. If the LR test from nnSVG to assess the statistical significance of SVGs is well calibrated, we expected to see a flat, uniform distribution representing null tests for most of the distribution with a spike close to 0 representing the non-null tests. When we apply filtering to remove lowly expressed genes (default for nnSVG and the main results presented in this work), we found approximately uniform distributions of $p$-values with additional spikes near 0 and 1 for the three datasets Visium human DLPFC, ST mouse OB, and Slide-seqV2 mouse HPC, respectively (Supplementary Fig. S19A–C). The spike near 0 represents the subset of significant SVGs in each dataset, as expected, while the spike near 1 suggests that the $p$-values for non-spatially-correlated genes may be somewhat too conservative overall, giving more values near 1 than expected. In particular, we observe that the spike near 1 is much larger when running nnSVG without filtering low-expressed genes (Supplementary Fig. S19D–F), indicating that many of the genes with $p$-values near 1 are low-expressed genes. In this way, while the process of filtering lowly expressed genes can lead to some false negatives (depending on the stringency of filtering), overall, we view this filtering step as helpful as nnSVG still recovers hundreds to thousands of significant SVGs in each dataset after filtering, and the rankings of the top SVGs are relatively unaffected by the stringency of the filtering since these genes tend to be highly expressed (Fig. 1e).

## nnSVG scales linearly with the number of spatial locations

Here, we illustrate how the computational complexity and runtime of nnSVG scales linearly with the number of spatial locations, which is crucial for identifying SVGs in datasets with thousands or more spatial locations. To demonstrate the linear scalability of nnSVG, we generated simulations by subsampling the numbers of spots ($n$ = 200, 500, 1000, 2000, 3639 in Visium human DLPFC, $n$ = 1000, 2000, 5000, 10,000, 20,000, 40,000, 53,208 in Slide-seqV2 mouse HPC, where the maximum number in each case is the full number of spots available in the dataset, including low-quality and unannotated spots). We ran nnSVG 10 times for a single gene at each number of spots using a single processor core and recorded runtimes, demonstrating a clear linear trend in the runtimes (Fig. 3a, b). We also compared the scalability of nnSVG for the Slide-seqV2 mouse HPC dataset with and without covariates for spatial domains included, and observed only a minimal increase in runtimes with covariates included (Supplementary Fig. S20A, B). Next, we compared the scalability of both nnSVG and SPARK-X against the earlier cubically scaling methods, SpatialDE[12,42] and SPARK[25], by subsampling spots from the Visium human DLPFC dataset and running each method 10 times for two genes (SPARK and SPARK-X require at least two genes to run without error) using a single core, which clearly demonstrated the cubic scaling of SpatialDE and SPARK (Supplementary Fig. S20C).

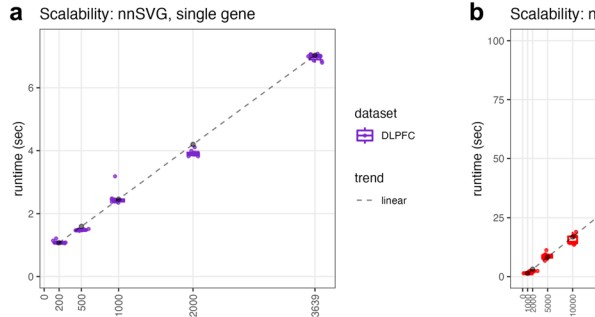

**Fig. 3 | nnSVG scales linearly with the number of spatial locations.** The runtime in seconds (*y*-axis) of nnSVG on a single gene (run *n* = 10 times on a single processor core) using downsampled numbers of spots (*x*-axis) with two transcriptome-wide datasets, **a** Visium human DLPFC and **b** Slide-seqV2 mouse HPC, without quality control or filtering of spots. The dashed lines represent linear trends in scalability for each dataset. Boxplots show medians, first and third quartiles, and whiskers extending to the furthest values no more than 1.5 times the interquartile range from each quartile.

Finally, we recorded runtimes for each of the three transcriptome-wide SRT datasets presented in this work (while filtering for lowly expressed genes) using both nnSVG and SPARK-X. We recorded runtimes of 520 s, 2788 s (46 min), and 18,642 s (5 h) for nnSVG for the 3 datasets (ST mouse OB, Visium human DLPFC, Slide-seqV2 mouse HPC), which contained 260, 3582, and 15,003 spots, respectively (after quality control and retaining annotated spots only). This compared to 11, 34, and 119 s for SPARK-X, respectively (Supplementary Fig. S20D). We used 10 processor cores on a high-performance compute cluster for all datasets for nnSVG and the Slide-seqV2 mouse HPC dataset for SPARK-X, and 1 core for the ST mouse OB and Visium human DLPFC datasets for SPARK-X due to the higher efficiency of SPARK-X.

### Deviance residuals from binomial model for baseline method and preprocessing
As an alternative nonspatial baseline instead of HVGs, we also considered deviance residuals from a binomial model, which has been shown to give an improved ranking of genes in the context of scRNA-seq data and is more theoretically justified due to the use of a count-based model[20]. We evaluated the binomial deviance residuals baseline by comparing the rank order of the selected SVGs in the main results against HVGs for each dataset, and found that while the individual rankings changed for some genes, in particular *NPY* in the Visium human DLPFC dataset and *Rgs14* in the Slide-seqV2 mouse HPC dataset, the overall performance and relative ranking of methods was similar to HVGs (Supplementary Fig. S21).

We also evaluated the results from nnSVG using deviance residuals from a binomial model for preprocessing, instead of log-transformed normalized counts (logcounts) for preprocessing[17] used in the main results, which has been shown to give improved performance in the context of scRNA-seq data[20]. We compared the rank order of the selected SVGs from nnSVG using the two preprocessing methods for each dataset, and found that the overall ranking of methods was similar (Supplementary Fig. S22).

### Applying nnSVG to datasets with multiple samples
The nnSVG model has been developed for data from one sample (tissue section) at a time. To evaluate the stability of the rankings of SVGs obtained from multiple samples, we applied nnSVG to each of the additional samples available in the original source for the Visium human DLPFC dataset (12 samples from 3 donors)[8,43]. We calculated the Spearman correlation between the rankings from each pair of samples and found that these correlations were relatively high (> 0.8) between samples within donors 1 and 3, respectively, moderate (> 0.75) between samples between donors 1 and 3, and lower within donor 2 and between donor 2 and the other donors (Supplementary Fig. S23A). This reflects the known biological structure from prior

analyses of this dataset[8], which found that the samples from donors 1 and 3 had all cortical layers present, while the samples from donor 2 were missing several cortical layers (Supplementary Fig. S23B; donors in rows). We also visualized the rank comparisons for the samples with the highest and lowest correlations with sample 151673 (the sample used in the main results) (Supplementary Fig. S23C, D) to further demonstrate these results.

In order to apply nnSVG to datasets with multiple samples in practice, we have also developed an approach based on averaging the ranks of the SVGs identified within each sample, which has been successfully applied in a new dataset[44] and is described in detail in the package documentation (vignette).

## Discussion
We have introduced nnSVG, a method to identify spatially variable genes (SVGs) in SRT data based on statistical advances in computationally scalable parameter estimation in spatial covariance functions in Gaussian processes using nearest-neighbor Gaussian process (NNGP) models[30,31]. In summary, our method (i) identifies genes that vary in expression continuously across the entire tissue or within a priori defined spatial domains, (ii) uses gene-specific estimates of length scale parameters within the Gaussian process models, and (iii) scales linearly with the number of spatial locations (Table 1). We have demonstrated the importance of fitting gene-specific length scale parameters within the GP models in application of SRT datasets to identify genes with different spatial ranges in their expression patterns within the tissue of interest. The linear scalability aspect is crucial for current technological platforms with thousands of spatial locations per tissue sample and for emerging platforms at even higher resolutions, such as 10x Genomics Visium HD. Compared to existing methods, while the runtime for SPARK-X[27] is fast, this method fits a fixed combination of covariance functions and length scale parameters across all genes, thereby leading to reduced flexibility to identify SVGs with different spatial ranges in expression. While earlier methods such as SpatialDE[12] fit gene-specific length scale parameters, these do not scale linearly with the number of spatial locations. The importance of fitting gene-specific length scale parameters is likely to represent a general finding for the analysis of SRT datasets, with applicability beyond the specific modeling approach used here.

Furthermore, unlike previous studies introducing methods to identify SVGs[12,27], we comprehensively evaluated our method against baseline methods. We compared against HVGs, deviance residuals from a binomial model[20], and Moran's I statistic[21], representing both non-spatial and spatial baseline methods, to assess the advantage in terms of performance of applying our more statistically sophisticated (and more computationally intensive) approach instead of relying on simpler baseline methods. We demonstrated the degree of overlap between the

**Table 1 | Summary of characteristics of methods included in the performance evaluations and runtime comparisons**

| Method | Spatial information | Flexible length scale parameters | Covariates for spatial domains | Runtime |
|---|---|---|---|---|
| nnSVG | ● | ● | ● | ◐ |
| SPARK-X | ● | ○ | ● | ● |
| HVGs | ○ | ○ | ○ | ● |
| Moran's I | ● | ○ | ○ | ◐ |
| SpatialDE | ● | ● | ● | ○ |
| SPARK | ● | ● | ● | ○ |

For each method, the columns indicate whether (●) or not (○) the method: (i) takes spatial information into account, (ii) fits models with flexible gene-specific length scale parameters, (iii) provides an option to include covariates for spatial domains in the models, and (iv) provides fast runtimes. Half-filled circles (◐) indicate intermediate scores. The scalable methods are shown in the first 4 rows, and the earlier cubically scaling methods (SpatialDE and SPARK) are shown in the last 2 rows.

nonspatial HVGs and nnSVG in each dataset, with a higher overlap expected in datasets where the biologically informative SVGs are largely related to spatial distributions of cell types. In general, our baseline comparisons demonstrate that HVGs provides excellent performance and computational efficiency in many datasets (despite not using the spatial information directly), especially where the spatial expression patterns are largely due to spatially distributed cell types—while nnSVG provides further improved performance in certain datasets with more complex expression patterns at higher computational cost.

We envision two types of primary applications of nnSVG. First, nnSVG can be used to generate lists of top SVGs during exploratory unsupervised analyses of SRT datasets, with the aim of detecting possible markers of biological processes of interest for further experimental validation. For example, ref. 8 applied this strategy using SpatialDE[12], using extensive computational resources due to the cubic scaling of this method[8]. These analyses are more feasible with nnSVG than with existing scalable methods (e.g., SPARK-X[27]), since nnSVG fits a gene-specific length scale parameter while also achieving linear computational scalability. For these types of analyses, the user can either select an arbitrary set of top-ranked SVGs (e.g., top 100 genes) or select a set of statistically significant SVGs with adjusted $p$-values from the LR test.

The second application of nnSVG is to use the set of top-ranked SVGs as the input for further downstream analyses, such as spatially-aware unsupervised clustering, for example[8,13,14], or registering the spatial locations of scRNA-seq data[11,15,16]. This type of analysis is analogous to standard workflows for scRNA-seq analyses[17]. In spatial data, we can modify this workflow by replacing the set of top HVGs with the set of top SVGs from nnSVG, and then perform unsupervised clustering on the set of top SVGs. Since the set of SVGs has been generated by methodology that takes spatial information into account, this gives a spatially-aware clustering of cell populations[8,40]. Our results demonstrated improved performance compared to using (nonspatial) HVGs for clustering, consistent with previous results[40].

Our method has some limitations, and we have identified several open directions for future work to extend our approach. First, while our method scales linearly with the number of spatial locations, the computational requirements remain nontrivial. For transcriptome-wide datasets with ≥10,000 spatial locations, runtimes are on the order of several hours when using 10 processor cores on a high-performance compute cluster. Since runtime depends on the number of genes, this can be reduced with more stringent gene filtering. In addition, our implementation is parallelized, allowing the user to select more cores if available, which will reduce runtimes. Future work could aim to further improve runtimes for large datasets, for example using low-rank statistical models that smooth the data into a smaller number of knots or inducing points representing the spatial locations, or further computational optimizations. Second, we observe some small

negative values in the estimated LR statistics, which are difficult to interpret. Since this occurs mainly for lower-ranked genes, this does not affect the rankings in the sets of top-ranked SVGs. This could be improved by developing adaptive filtering thresholds that carefully remove low-expressed genes, which could also improve the calibration of $p$-values for low-expressed genes. Similarly, constraints could be placed on low values of the estimated length scale parameter within the models, although we found that these were generally low-expressed genes that were either filtered out or were not ranked as top SVGs. Third, while nnSVG identifies individual SVGs, we have not grouped these into gene groups or metagenes. Future work could develop added functionality to group genes into biologically interpretable metagenes in an unsupervised manner, similar to refs. 12,27. Fourth, our model has been developed for a single sample (tissue section) at a time. While we have implemented a practical approach to apply nnSVG to multiple-sample datasets based on averaging the ranks of the SVGs identified within each sample, future work could focus on developing a principled statistical approach for multiple-sample datasets, for example by jointly estimating parameters across multiple samples to improve power and robustness. Finally, while we calculate an effect size defined as the proportion of spatial variance (similar to ref. 12), this definition does not distinguish between technical and biological variance, in contrast to standard effect size definitions in scRNA-seq workflows[17] (Supplementary Fig. S24). Future work could aim to define a modified effect size that decomposes total variance into technical and biological components as well as spatial and nonspatial components, e.g., using a concept of biological spatial variance, which would aid in the interpretation of top-ranked SVGs.

Our method is implemented as an R package within the Bioconductor framework[34], and is freely available from Bioconductor at https://bioconductor.org/packages/nnSVG.

## Methods
### Preprocessing
The nnSVG workflow begins with preprocessing steps. For the analyses in this manuscript, we applied standard quality control (QC) to each dataset to filter out low-quality spatial locations (spots), using functions to calculate QC metrics implemented in the `scater`[45] R/Bioconductor package. The thresholds we used for each QC metric can be found in our code repository (see "Code availability").

Next, we filter out low-expressed genes and mitochondrial genes. Low-expressed genes are assumed to largely represent noise and to be unlikely to provide significant biological information about spatially-resolved biological processes, so removing them improves computational performance while preserving most of the information. For the analyses in this manuscript, we used the following filtering thresholds. For the Visium human DLPFC dataset, we retained genes with at least 3 unique molecular identifier (UMI) counts in at least 0.5 percent of spatial locations. For the Slide-seqV2 mouse HPC dataset, we retained genes with at least 1 UMI count in at least 1 percent of spatial locations. For the ST mouse OB dataset, we retained genes with at least 5 UMI counts in at least 1 percent of spatial locations. For the seqFISH mouse embryo dataset, no filtering was needed, as this dataset contains a smaller set of targeted genes. By contrast, mitochondrial genes are observed to be very highly expressed in most single-cell datasets, but their expression is generally not considered to be informative for distinguishing cell populations or states, so removing them reduces noise[17]. For the analyses in this manuscript, we removed mitochondrial genes from all datasets. The nnSVG package provides a filtering function for both low-expressed and mitochondrial genes, with default values appropriate for the 10x Genomics Visium platform, which can also be adjusted or disabled by the user.

Next, we normalize and transform the raw UMI counts using the log-transformed normalized counts methodology (also referred to as logcounts) using library size factors implemented in

the `scran`, `scuttle`, and `scater` R/Bioconductor packages[45,46]. Normalization reduces technical biases between measurements from different spots, while log-transformation transforms the counts to a continuous and approximately normally distributed scale, allowing the NNGP models to be fitted. As an alternative to logcounts, we also demonstrate the use of the binomial deviance residuals methodology implemented in the `scry` R/Bioconductor package[20], which has been shown to give improved performance in scRNA-seq data[20].

## nnSVG model and parameters

In the nnSVG methodology, we assume that the input data consists of preprocessed gene expression measurements for thousands of genes at a set of spatial locations on a tissue slide, with the spatial locations typically also numbering in the thousands. The core of the nnSVG methodology consists of fitting a nearest-neighbor Gaussian process (NNGP) model[30,31] to the preprocessed expression measurements for each gene, i.e., one model per gene. This model is defined as:

$$\mathbf{y} \sim N(\mathbf{X}\boldsymbol{\beta}, \widetilde{\Sigma}(\boldsymbol{\theta}, \tau^2)) \tag{3}$$

Here, $\mathbf{y} = (y_1, ..., y_N)$ represents a vector of normalized and transformed expression values for gene $g$ (omitting the index $g = 1, ..., G$ for simplicity) at a set of spatial locations $\mathbf{s} = (\mathbf{s_1}, ..., \mathbf{s_N})$. The spatial locations are assumed to be two-dimensional, but may in principle be generalized to higher dimensions. The $\widetilde{\Sigma}(\boldsymbol{\theta}, \tau^2)$ term represents the NNGP covariance matrix, which provides a scalable (in linear-time and storage) approximation to the covariance matrix $\Sigma(\boldsymbol{\theta}, \tau^2) = C(\boldsymbol{\theta}) + \tau^2 \mathbf{I}$ from a full GP model, which scales cubically in the number of spatial locations. The GP covariance matrix $C(\boldsymbol{\theta}) = (C_{ij}(\boldsymbol{\theta}))$ (also referred to as a kernel) captures the spatially correlated variation and is parameterized by a vector of parameters $\boldsymbol{\theta}$. We assume an exponential covariance function, based on the observation that the widely used squared exponential function (e.g., used previously in SpatialDE[12]) decays too rapidly with distance in the context of SRT data[36]. The exponential covariance function (or kernel) is defined as:

$$C_{ij}(\boldsymbol{\theta}) = k(\mathbf{s_i}, \mathbf{s_j}) = \sigma^2 \exp\left(\frac{-||\mathbf{s_i} - \mathbf{s_j}||}{l}\right) \tag{4}$$

with covariance parameters $\boldsymbol{\theta} = (\sigma^2, l)$, and where $||\mathbf{s_i} - \mathbf{s_j}||$ represents the Euclidean distance between two spatial locations $\mathbf{s_i}$ and $\mathbf{s_j}$. In this parameterization, $\sigma^2$ represents the spatial component of variance, and $l$ is referred to as the length scale (or bandwidth) parameter, which controls the strength of decay of correlation with distance. The final parameter $\tau^2$ in equation (3) is referred to as the nugget, which represents the additional nonspatial component of variance.

Alternatively, the Gaussian process model $\mathbf{y} \sim N(\mathbf{X}\boldsymbol{\beta}, C(\boldsymbol{\theta}) + \tau^2 \mathbf{I})$ may also be written as:

$$y(\mathbf{s}) = m_{\boldsymbol{\theta}}(\mathbf{s}) + w(\mathbf{s}) + \epsilon(\mathbf{s}), \qquad \epsilon(\mathbf{s}) \overset{iid}{\sim} N(0, \tau^2) \tag{5}$$

where $w(\mathbf{s})$ follows a Gaussian process, $w(\mathbf{s}) \sim GP(0, C_{\boldsymbol{\theta}}(.,.))$, and $m_{\boldsymbol{\theta}}(\mathbf{s}) = \mathbf{x}(\mathbf{s})^T \boldsymbol{\beta}$.

In most applications of nnSVG, we assume an intercept-only model, where $\mathbf{X} = \mathbf{X}_{[N \times 1]} = \mathbf{1}_{[N \times 1]}$ and $\boldsymbol{\beta}$ accounts for the mean expression level. In this case, we are interested in identifying genes with any statistically significant spatial correlation in expression.

However, in some datasets, we are also interested in identifying SVGs within spatial domains, i.e., regions of the tissue slide corresponding to anatomical features or tissue types, which have been defined a priori, for example using morphology from histology images, or alternatively using unsupervised clustering. The nnSVG methodology facilitates these types of analyses by allowing the user to provide $\mathbf{X}$ as an $\mathbf{X}_{[N \times d]}$ design matrix containing up to $d - 1$ covariates, with covariate columns consisting of indicator variables for the spatial domains at each spatial location, or other known values per spatial location.

Our key parameter of interest in the model is $\sigma^2$. We perform model fitting and parameter estimation using the fast optimization algorithms for NNGP models implemented in the BRISC R package[32], which we use to obtain maximum likelihood parameter estimates for the covariance parameters $\boldsymbol{\theta} = (\sigma^2, l)$ and $\tau^2$, as well as the log-likelihoods of the fitted models. The computational complexity of the model fitting is $\mathcal{O}(n * m^3)$, where $n$ = number of spatial locations, $m$ = number of nearest neighbors, and the initial steps of ordering coordinates and calculating nearest neighbors are performed once only and are re-used for all genes. Note that BRISC also provides the option to obtain precise bootstrap estimates for the parameter estimates, which we do not use here, due to the computational tradeoff when fitting thousands of models (one model per gene) for SRT data.

Within the BRISC algorithm[32], we use the parameter choices `order` = "AMMD" (approximate maximum minimum distance ordering of coordinates, see ref. 47 for details) and `n.neighbors` = 10 (10 nearest neighbors, which has been shown to retain a large proportion of information[30]) as default values, while also allowing the user to adjust these choices. Additional details are provided in the nnSVG and BRISC package documentation.

Next, we perform inference on the estimated $\sigma^2$ parameters per gene, where we test $H_0$: $\sigma^2 = 0$ vs. $H_1$: $\sigma^2 > 0$. We use a likelihood ratio test (LR) for the inference, where we compare the log-likelihood of the fitted model against a classical linear model that assumes $\sigma^2 = 0$ and hence does not account for spatial correlation in the data. We use the estimated LR statistics to generate an overall ranking of SVGs in terms of the strength of their spatial expression patterns. We also calculate approximate $p$-values for statistical significance per gene using an asymptotic $\chi^2$ distribution with two degrees of freedom (since there are 2 fewer parameters, $\boldsymbol{\theta} = (\sigma^2, l)$ in the simpler model) and apply the Benjamini-Hochberg method[48] to adjust the $p$-values for multiple testing across genes. The user can then select either (i) an arbitrary number of top-ranked SVGs (e.g., top 100 or 1000) for further investigation or to use as the input for downstream analysis methods, analogous to scRNA-seq workflows[17], or (ii) a set of statistically significant SVGs by applying a threshold (e.g., 0.05) to the multiple testing adjusted $p$-values.

Since the nnSVG methodology fits a separate model for each gene, the length scale parameter $l$ is estimated individually per gene. This flexibility is the most important reason explaining the improved performance of nnSVG compared to other scalable methods, since the gene-specific length scale parameter allows nnSVG to identify SVGs from distinct biological processes with different spatial ranges in expression within the same tissue slide.

Finally, we also calculate an estimated effect size per gene, defined as the proportion of spatial variance (out of total variance), i.e., the proportion of variance explained by spatial dependencies, as previously defined by ref. 12:

$$\text{propSV} = \frac{\sigma^2}{\sigma^2 + \tau^2} \tag{6}$$

## Computational implementation

nnSVG is implemented as an R package within the Bioconductor[34] framework, using the BRISC R package[32] for model fitting and parameter estimation, and the BiocParallel R package[49] for parallelization. We extended the BRISC package (version 1.0.4) to apply these methods to SRT data, in particular, to extract the fitted log-likelihoods (for the LR tests) and to improve runtime when fitting thousands of models (one per gene) by re-using the ordering of spatial coordinates and calculating nearest neighbors.

The nnSVG package re-uses existing infrastructure for scRNA-seq and SRT data within the Bioconductor framework[17,35], e.g., the `SpatialExperiment` object structure[35] to load input data and store results, which streamlines integration into existing Bioconductor-based analysis workflows.

## Visium human DLPFC dataset

The Visium human DLPFC dataset consists of a single sample of human brain tissue from the dorsolateral prefrontal cortex (DLPFC) region, measured with the 10x Genomics Visium platform[50]. This dataset was published by Maynard et al.[8] and previously released through the `spatialLIBD` R/Bioconductor package[43]. The Visium platform measures transcriptome-wide gene expression at a hexagonal grid of spatial locations (referred to as spots) on a tissue slide, with overall dimensions 6.5 mm × 6.5 mm, spots 55 μm in diameter, and 100 μm between spot centers[50]. The dataset used here consists of one biological sample (sample 151673) from one donor, out of the 12 samples (3 donors) in the original study by Maynard et al.[8]. This sample contains transcriptome-wide gene expression measurements at 3639 spots overlapping with the tissue area. We use all 12 samples for the additional multiple-sample analyses. In the original study, spots were manually annotated with labels for the six cortical layers and white matter[8], which we use as approximate ground truth labels for method evaluation.

## Slide-seqV2 mouse HPC dataset

The Slide-seqV2 mouse HPC dataset consists of gene expression measurements in a tissue sample from the mouse hippocampus (HPC), measured with the Slide-seqV2[4] platform and published by Stickels et al.[4]. Spot-level annotations for cell types were generated computationally by Cable et al.[39], which we use here to define spatial domains representing anatomical regions within the hippocampus (in particular the region defined by CA3 cell type labels). This dataset consists of a total of 53,208 spatial locations (referred to as beads for this platform), 15,003 of which have been annotated with cell type labels by Cable et al.[39]. In the analyses of this dataset, we are especially interested in genes with spatial gradients of expression within the CA3 region of the hippocampus, which have previously been identified by Cable et al.[39].

## ST mouse OB dataset

The ST mouse OB dataset was generated by Ståhl et al.[1], consisting of gene expression measurements in the olfactory bulb (OB) region of the mouse brain. This technological platform (Spatial Transcriptomics) was subsequently further developed (e.g., to increase resolution and simplify experimental procedures) by 10x Genomics as the Visium platform. Therefore, ST represents an earlier iteration of the 10x Genomics Visium platform. The ST mouse OB dataset consists of transcriptome-wide gene expression measurements at 260 spatial locations (referred to as spots) after quality control filtering, from a single sample from the original study[1], and has previously been re-analyzed in several studies including[12].

## seqFISH mouse embryo dataset

The seqFISH mouse embryo dataset consists of expression measurements of 351 targeted genes within a sagittal tissue section of a mouse embryo from a study investigating mouse organogenesis[38] using the seqFISH platform[33]. The seqFISH platform is a molecule-based SRT platform, which allows individual mRNA molecules to be identified at sub-cellular resolution. In the subset of the data used here, these measurements are summarized at single-cell resolution. The data used here consists of the cells from a single embryo and section (embryo 1, z-slice 2) from the original study[38].

## Baseline methods

We compared performance against the following baseline methods: (i) highly variable genes (HVGs)[17], (ii) deviance residuals under a binomial model[20], and (iii) Moran's I statistic[21].

HVGs are widely used in scRNA-seq analysis workflows, with implementations provided in the Bioconductor[17], Seurat[18], and Scanpy[19] frameworks. Here, we used the standard definition of HVGs from ref. 17 implemented in the `modelGeneVar()` function in the `scran` R/Bioconductor package[46]. In this definition, the HVGs methodology fits a mean-variance trend to the log-transformed normalized expression values (logcounts) per gene and ranks genes by excess biological variation, defined as the excess variance above the trend for each gene, under the assumption that the trend represents technical variance[17]. To apply HVGs to SRT data, we calculate the gene-specific means and variances by treating each spot as equivalent to a cell. This method does not make use of any spatial information.

The deviance residuals methodology assumes a binomial model (i.e., count-based instead of log-transforming to continuous values) and ranks genes by the deviance residuals from the fitted binomial models[20]. Compared to HVGs, this approach has been shown to give an improved ranking of genes in scRNA-seq data, and is more theoretically justified due to the use of a count-based model[20]. We apply this method to SRT data by treating each spot as equivalent to a cell. As for HVGs, this method does not make use of any spatial information.

Moran's I statistic[21] is a standard statistical measure of spatial autocorrelation, which can be calculated from the log-transformed normalized expression values for each gene. Values range from +1 (perfect spatial correlation) to 0 (no spatial correlation) to −1 (perfect spatial anticorrelation). In SRT data, the values for most genes are between 0 and 1, and negative values usually do not have a clear biological meaning. We use Moran's I statistic to rank genes as SVGs, with the highest values (close to +1) representing the top-ranked SVGs. The Moran's I formula requires an assumed weights matrix, which we calculate as the inverse squared Euclidean distances between spots, which is consistent with implementations provided in the Seurat workflow[18] and in the 10x Genomics Space Ranger/Loupe software (which also includes truncation at 36 neighbors)[51]. We use the `Rfast2` R package[52] to calculate the Moran's I statistic values.

## Reporting summary

Further information on research design is available in the Nature Portfolio Reporting Summary linked to this article.

## Data availability

The datasets used for the analyses in this manuscript can be downloaded in `SpatialExperiment` format[35] from the `STexampleData` Bioconductor package[53], which includes annotation labels from the original sources, and the `spatialLIBD` Bioconductor package[43]. The original datasets and annotations are sourced from refs. 8,43 (Visium human DLPFC dataset), refs. 4,39 (Slide-seqV2 mouse HPC dataset), ref. 1 (ST mouse OB dataset), and ref. 38 (seqFISH mouse embryo dataset). Source data files to reproduce figures in the manuscript are also available from Figshare at https://doi.org/10.6084/m9.figshare.23561439.v2. All other data supporting the findings of this study are available within the article and its supplementary files. Any additional requests for information can be directed to, and will be fulfilled by, the lead contact.

## Code availability

nnSVG is freely available as an R package from Bioconductor (as of 2023-06-14: nnSVG version 1.4.1 available in Bioconductor release version 3.17) at https://bioconductor.org/packages/nnSVG. The package is also available from GitHub at https://github.com/lmweber/nnSVG. Code to reproduce all preprocessing, analyses, and figures in this manuscript is available from GitHub at https://github.com/lmweber/nnSVG-analyses. An archived version of this code repository as of the time of publication is also available[54]. We used nnSVG version 1.3.10 for the analyses in this manuscript.

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

## Acknowledgements

We thank our collaborators at the Lieber Institute for Brain Development for input and feedback on the application of methods to identify SVGs in SRT data and ongoing collaborations which generated the ideas for the methods developed in this manuscript. We also thank the maintainers of the Joint High Performance Computing Exchange (JHPCE) compute cluster at Johns Hopkins Bloomberg School of Public Health for providing essential computing resources. Research reported in this publication was supported by the National Institute of Mental Health (NIMH) of the National Institutes of Health (NIH) under the award number U01MH122849 (S.C.H., L.M.W.), the National Human Genome Research Institute (NHGRI) of the NIH under the award number K99HG012229 (L.M.W.), the National Institute of Environmental Health Sciences (NIEHS) of the NIH under the award R01ES033739 (A.D.), National Science Foundation Division of Mathematical Sciences grant DMS-1915803 (A.D.), and CZF2019-002443 and CZF2018-183446 (S.C.H., K.D.H.) from the Chan Zuckerberg Initiative DAF, an advised fund of Silicon Valley Community Foundation.

## Author contributions

L.M.W. developed the nnSVG methodological framework using BRISC and LR tests for SRT data, implemented the nnSVG software package, performed analyses, created figures, and drafted text. A.S. extended the BRISC software package for use within the nnSVG framework, and provided input on the methodological framework, software implementation, and text. A.D. provided advice on the application of NNGP models to SRT data, and provided input on the methodological framework and text. K.D.H. provided input on the methodological framework, software implementation, analyses, figures, and text. S.C.H. supervised the project; provided input on the methodological framework, software implementation, analyses, figures, and text; and drafted text. All authors approved the final manuscript.

## Competing interests

The authors declare no competing interests.
