## [Peer review file · Nature Communications]

REVIEWER COMMENTS

Reviewer #1 (Remarks to the Author):

In this manuscript authors present a method (nnSVG) that propose to solve the problem of finding spatially variable genes (SVG). The problem of finding SVGs from the spatial transcriptomic(ST) data is a well-established problem with multiple methods already published however the key innovation of the proposed method is to find SVGs across varying spatial ranges within the same tissue, while also being computationally tractable. One can potentially design a brute force algorithm to solve such a variation of the problem but that would be computationally expensive, which makes the proposed method particularly exciting as the field of ST is advancing rapidly to profile hundreds of thousands of spots.

Overall, the manuscript is excellently written. Authors have gone out of their way to explain the difference in SVG and domain maker genes in the introduction, which is often a point of confusion for a reader and the efforts should be appreciated. Methods section is clearly explained, software tool is available through BioC and example dataset provided online also works as expected. I have a few minor concerns on the result section which are as the following:

* The idea of finding a better subset of SVG is exciting but in most of the single-cell analyses finding such list is not the sole goal, generally it's accompanied with spatially-(un)aware clustering. I understand the motivation of this paper is "not" to improve clustering but I think authors should at least comment on the performance of downstream analyses when provided with HVG, SVG, nnSVG. Does having nnSVG identifies a subgroup of cells not identifiable before ?

* Figure 1F: This result is very exciting but I have two comments here (Left figure) Spearman correlation of 0.78 is great however it's worth exploring some of the outliers for example there are bunch of genes with x-axis rank around 750 but y-axis of 250. I am curious if these non-diagonal genes are "only" spatially variable, do they have similar length scale parameter, more context is needed here. Also, I wonder if there is a better way to show this results for example plotting effect size "propSV" instead of individual variance. (Right figure) I think it's worth marking MOBP, PCP4, SNAP25 on the MoransI baseline, provided these are spatial markers and MoransI statistics finds spatial autocorrelation.

* Lastly, Figure 3 shows the performance of only nnSVG, given one of the advantage of using nnSVG over other method is speed, it's worth showing other methods on the plot, which surely will be very high comparatively.

Reviewer #2 (Remarks to the Author):

The authors propose an approach "nnSVGs" based on nearest-neighbor Gaussian processes to identify spatially variable genes (SVGs). The proposed approach is superior to the existing methods in that it can identify genes whose expression vary continuously across the entire tissue or within a spatial domain and it allows gene-specific estimates of length scale parameters within the Gaussian process models. Meanwhile, the computational and storage cost of nnSVGs is linear to the number of spatial locations. I find the application of NNGP in identifying SVGs to be interesting and useful. The paper is well-written in general. I have some comments about the technique details of the methodology and data analyses as detailed below.

1. The authors use the exponential covariance function as the kernel of the Gaussian process to model spatial correlation. The hyperparameters ($\sigma ²$, l) in the exponential covariance function are estimated by the maximum likelihood estimates using optimization. My concern here is that when

the length scale (parameter l) is very small, a Gaussian process with an exponential kernel can behave like a white noise process. In other words, the estimate of the spatial component of variance σ^2 can be inaccurate when the length scale is very small since the model can hardly distinguish the spatial variance from the white noise variance (nugget). Therefore, I think it will be reasonable to setup a lower bound for the length scale when finding the maximum likelihood estimates. According to Figure 1B, the estimated length scale can be very close to 0. It will be interesting to see whether a lower bound for l can improve the performance of nnSVG or not.

2. The data analyses in Section 2 imply that the proposed nnSVG outperforms SPARK-X but is in general inferior to the baseline method HVGs in recovering SVGs. Table 1 indicates that HVGs is less expensive than nnSVG. Though the authors point out that nnSVG is more flexible than HVGs, it is still not very clear to me why it is important to use nnSVG rather than HVGs. It will be better if the authors can highlight more on why and when one should use nnSVG instead of HVGs (and other methods).

Some notes about writing:

- 1) The sentence from line 173 to 176 is confusing.
- 2) Line 436-447 is exactly the same as line 88-89. A little rewrite might be better.

I hope the authors find these comments helpful for improving their manuscript.

Reviewer #3 (Remarks to the Author):

I've included all of my comments below.

Summary:

This manuscript outlines a new method, nnSVG, designed for the detection of spatially variable genes (SVGs) in spatially resolved transcriptomics data. The genes can exhibit spatial variance either across the whole tissue or within pre-annotated regions, depending on the design matrix used during analysis. The method relies on a well-established idea of using Gaussian Processes (GPs) to model gene expression (presented by Svensson et al. in 2018) and the use of a likelihood-ratio test (LR test) -- comparing with a linear model -- to determine whether spatial information is relevant to the expression. While relying on already existing concepts, the authors introduce new and relevant elements in their method; the most notable one being the use of a Nearest Neighbor Gaussian Process (NNGP) to obtain linear (w.r.t. observations) scaling rather than the cubic one observed in standard GP inference. They also use gene-specific estimates of the length scale parameter, something which should better account for differences in expression patterns between genes involved in disparate biological processes.

Comments:

I found this manuscript to be extremely well-written and pleasant to read, everything from structure to language was clear. Section 2.2 “*Key Innovations of nnSVG*” reminds me of similar sections often found in papers from the machine learning community, this is something I really appreciate and believe helps the reader understand what sets this method apart from already existing ones. My only complaint, with respect to the text, would be that section 2.1 is more or less identical to parts of the Methods section. While I personally like having the more technical aspects of a method as a part of the main text, it might not appeal to the average reader, and it could be wise to compress section 2.1 and refer to the Methods section for more details. Having given the authors some praise, there are also some concerns and comments that I’d like to share, these will be given below.

Major comments

1. **Impact:** While I like the technical innovation of this method, I’m less convinced of its impact. To elaborate, in most of the analyses it seems as if nnSVG produces similar results to what the authors refer to as *baseline methods*, but with a much increased computational cost. Unless I’ve misunderstood parts of the manuscript, I find no compelling evidence that simply using the variance or other simpler metrics to obtain non-spatial HVG genes wouldn’t be enough for a successful downstream analysis.

I realize this comment rather critiques the idea of SVGs than the authors’ method, which might seem unfair given how plenty of other methods have been published on the topic. However, as the topic becomes more saturated, I feel as if the need to really show *why* such methods are necessary also increases. Furthermore, when it comes to the application of finding spatially variable genes within certain domains, one could argue that something similar could be done using the variance, where the observations included in the variance calculation are restricted to those within a certain region of interest. Analyses that the authors could make in order to make a more convincing case would be, but are not limited to:

- a. Exemplifying how there’s a qualitative difference in the downstream results when using genes from nnSVG compared to HVGs, preferably to the benefit of the former.
 - b. Show that several essential genes, and not only a handful, are lost if only using HVGs
2. **False positives:** Another concern of mine is that of **false positives**. While it seems as if nnSVG is able to identify genes with spatial variance, it feels as if it’s fairly promiscuous in some analyses: for example, in the DLPFC data set 60% of the genes are considered as having a statistically significant spatial variance. I

understand the difficulty of reporting false positives for this kind of task, as there is no proper ground truth. However, I would maybe suggest the authors generate more complex synthetic data (not just a shuffling of the spots, which breaks *all* spatial structure). Maybe some synthetic data where the generative model is that of a GP could be used, one idea here would be to induce spatial dependence in only one or a few regions to see how the method handles this (not having any prior annotations of these methods).

3. **Synthetic data:** Continuing on the topic of synthetic data, in contrast to the full permutation study described in section 2.5.2. It would be interesting to see whether the method ranks genes in the correct order. For this purpose, I'd suggest an ablation study similar to what is described in: <https://academic.oup.com/bioinformatics/article/37/17/2644/6168120> (see Figure 1).
4. **Multiple sections:** Since SRT methods are becoming ever more popular, there's a trend toward generating larger data sets. Alas, the mouse OB section that the authors use stems from a larger set of 12 sections. My point is that these days, it's rare that we analyze a single section in isolation. I, therefore, have two questions pertaining to this matter:
 - a. How are multiple sections analyzed, is it done one-by-one or jointly? When using non-spatial metrics to identify variable genes, this is a trivial question (as observations are treated as independent no matter if they belong to the same or different tissue sections). I could envision a joint analysis where the entries between observations from non-same sections are nulled, but this also becomes somewhat untrue if these are adjacent sections or if they represent the same region. In a one-by-one analysis, I guess one could take the union (or intersection) of all the SVG genes. Another idea is to transfer the expression to a common coordinate framework (CCF), or in the case of highly similar sections simply align them and create a "consensus slide", to then apply nnSVG in the shared space. Either way, I would encourage the authors to elaborate on **multi-section analysis** and how their method handles this.
 - b. If the sections in a larger data set are analyzed one by one, how **robust** are these results? For example, if the authors were to analyze all 12 mouse OB sections or all the sections in the DLPCFC data set, how consistent would the results and ranks be? This could, for example, be evaluated with Kendall's Tau. Actually, *this question of robustness is something that I'd like the authors to address no matter the answer to a).*
5. **Run-times:** I would also like to see run-time estimates when using a more complex design matrix than simply an intercept. I expect this to increase the run-time somewhat as more parameters need to be fitted.
6. **Run-times:** Initially, upon reading the Introduction and Abstract, I thought that this manuscript would present some stellar run-times, as the authors obviously have put a lot of thought into this topic. However, I was somewhat negatively surprised to see the -- apologies for the expression, but it's the only true assessment -- poor run-time results. The nnSVG method takes on average 133x longer time to analyze the data compared to SPARK-X. HVG and SVG identification is usually given little room in SRT studies, and the main purpose of this task is usually to select genes for downstream analysis; I find it somewhat hard to justify a 5h wait just to get a list of spatially variable genes, this while also requiring 10 cores (which many personal computers do not even have). This makes me disagree with Table 1, where nnSVG is reported to have "intermediate runtimes", if only comparing with the 4 listed methods, I would definitely put this value as "not"; however if an older method like SpatialDE was included in the comparison, I could agree with the current "intermediate" classification. Could the authors either:
 - a. Motivate how this run-time is acceptable, especially in the context of what is gained from having these genes compared to standard HVG genes (relating to my question above)
 - b. Improve the run-time, I'm not familiar with all of the assumptions in the NNGP model, but could you perhaps use inducing points? Looking at the referenced paper, it seems as if Gibbs sampling

is used (though I'm not sure about the implementation), maybe this could be replaced by a more efficient algorithm that could be sped up by GPU acceleration?

Minor comments

1. In their abstract, the authors write "*to identify spatially variable genes is a key step during analyses of spatially resolved transcriptomics data*". While this might be true in the future, I'm keen to argue that such is not the case at the present time. Most SRT-based studies settle with just HVGs and do not seem to rely on SVGs.
2. On line 21 the authors write "*up to thousands of genes at near- or sub-cellular resolution*". To me "up to thousands" does not really convey the fact that the full transcriptome (or at least polyadenylated transcriptome) can be captured in many methods, maybe reconsider this choice of wording.
3. In most existing methods for detection of SVGs there seems to be a preference for ranking by adjusted p-value, is there any particular reason that the authors chose to rank their genes by the LR statistic?
4. In section 2.3.1 it's reported that out 3396 genes included in the analysis, 2049 were considered as SVG, and subsequently, 133 of the 134 known cortical layer-associated SVGs were correctly identified. Maybe the authors could provide a p-value for how likely this is to happen by chance; that is if I select 2049 genes from 3396 ones, how likely am I to get 133 "correct ones" out of 134. Fisher's exact test would probably be befitting for this task.
5. On lines 212-213 it's stated that the OB mouse data set contains a smaller set of spatial locations; since not everyone is familiar with the old ST platform, it might be beneficial to also mention that, in addition to this, the resolution is lower than the more familiar Visium platform.
6. On lines 260-261, concerning the Slide-seqV2 analysis, it's stated that HVG baseline methods do not perform as well as in the Visium data set. Is this statement solely based on whether the two genes were included in the set of the top 1000 most highly ranked ones? If that is true, I think that's a bit of a drastic conclusion, and I feel as if the ability to correctly rank genes should be assessed using a larger gene set.
7. It would be great if the authors reported the improved scaling properties when exchanging the standard GP for an NNGP. If I read the material correctly, it shifts from $O(n^3)$ to $O((n+k)*m^3)$ where n = number of observations, k = number of reference points, m = number of neighbors.

Response to Reviewer 1

In this manuscript authors present a method (nnSVG) that propose to solve the problem of finding spatially variable genes (SVG). The problem of finding SVGs from the spatial transcriptomic (ST) data is a well-established problem with multiple methods already published however the key innovation of the proposed method is to find SVGs across varying spatial ranges within the same tissue, while also being computationally tractable. One can potentially design a brute force algorithm to solve such a variation of the problem but that would be computationally expensive, which makes the proposed method particularly exciting as the field of ST is advancing rapidly to profile hundreds of thousands of spots.

Overall, the manuscript is excellently written. Authors have gone out of their way to explain the difference in SVG and domain maker genes in the introduction, which is often a point of confusion for a reader and the efforts should be appreciated. Methods section is clearly explained, software tool is available through BioC and example dataset provided online also works as expected. I have a few minor concerns on the result section which are as the following:

- The idea of finding a better subset of SVG is exciting but in most of the single-cell analyses finding such list is not the sole goal, generally it's accompanied with spatially-(un)aware clustering. I understand the motivation of this paper is "not" to improve clustering but I think authors should at least comment on the performance of downstream analyses when provided with HVG, SVG, nnSVG. Does having nnSVG identifies a subgroup of cells not identifiable before ?

We agree with the suggestion that some users will be interested in using the list of spatially variable genes (SVGs) from nnSVG as the input genes for downstream clustering. In this case, since the genes have been selected in a spatially-aware manner (i.e. taking spatial information into account), the downstream clustering should also be spatially-aware and, in theory, should outperform non-spatial clustering. We have included additional analyses in the revised manuscript to evaluate the improvement in clustering performance in this case. Specifically, for each method, we performed clustering on the top 50 principal components (PCs) calculated on either the top 1000 SVGs (for spatial methods) or the top 1000 highly variable genes (HVGs) (non-spatial), for the Visium human DLPFC dataset (**Supplementary Figure S15A-D**). We then evaluated clustering performance in terms of the adjusted Rand index (ARI) (which measures the similarity between two sets of cluster labels, with values ranging from 0 to 1) compared to the manually annotated cortical layer labels in this dataset (**Supplementary Figure S15E**). This showed that nnSVG and Moran's I (both of which are spatial methods) outperformed HVGs (non-spatial baseline method), as expected. In addition, nnSVG and Moran's I outperformed SPARK-X (alternative spatial method), which is consistent with the main results showing that the top SVGs from nnSVG more closely reflect the biological structure in this dataset, compared to SPARK-X (**Figure 1C** and **Supplementary Figures S5-S6**).

In addition, we note that a recent benchmarking paper also demonstrated that the use of top SVGs as the input for clustering gave improved downstream clustering performance (compared to using HVGs) in datasets from several platforms (Li et al., 2022), which is consistent with our results. We have added a citation to this study in the Discussion in our manuscript.

New Figure: Supplementary Figure S15. (Reproduced from manuscript.)

- Figure 1F: This result is very exciting but I have two comments here (Left figure) Spearman correlation of 0.78 is great however it's worth exploring some of the outliers for example there are bunch of genes with x-axis rank around 750 but y-axis of 250. I am curious if these non-diagonal genes are "only" spatially variable, do they have similar length scale parameter, more context is needed here. Also, I wonder if there is a better way to show this results for example plotting effect size "propSV" instead of individual variance. (Right figure) I think it's worth marking MOBP, PCP4, SNAP25 on the MoransI baseline, provided these are spatial markers and MoransI statistics finds spatial autocorrelation.

Thank you for these suggestions. We have investigated these findings in more detail by overlaying the length scale and effect size parameter estimates respectively onto the rank comparisons (**Supplementary Figure S3A-C**). Overall, this shows that in the comparison with HVGs, the off-diagonal genes in **Figure 1F** (left panel) with high x-axis rank and low y-axis rank tend to have relatively large length scales (**Supplementary Figure S3A-B**, highlighting genes with length scale less than or greater than 0.15 respectively) together with relatively high effect sizes (**Supplementary Figure S3C**). These genes include *CALM1* and *CST3* (see labels in **Supplementary Figure S3C**, left panel, red text), which have relatively high expression across the whole tissue section (**Supplementary Figure S3D**, top row), with some spatial variation between the white matter and gray matter regions and at the boundary between the white matter and gray matter. By contrast, in the comparison with Moran's I, the off-diagonal genes in **Figure 1F** (right panel) have relatively small length scales (**Supplementary Figure S3A-B**) together with high effect sizes (**Supplementary Figure S3C**). These genes include *IGHM* and *HBA2* (labels in **Supplementary Figure S3C**, right panel, red text), which have relatively low expression that matches the sparse expression patterns for the immune and blood-related genes from **Figure 1A** (**Supplementary Figure S3D**, bottom row). Overall, these results

demonstrate that nnSVG identifies some SVGs that are not captured adequately by either HVGs or Moran's I.

We have also labeled the additional genes (*MOBP*, *PCP4*, *SNAP25*) in **Figure 1F** (right panel) in the main figures, as suggested -- which shows that these are highly ranked by both Moran's I and nnSVG.

New Figure: Supplementary Figure S3. (Reproduced from manuscript.)

- Lastly, Figure 3 shows the performance of only nnSVG, given one of the advantage of using nnSVG over other method is speed, it's worth showing other methods on the plot, which surely will be very high comparatively.

We have included additional evaluations to compare the computational scalability for the different methods, including the earlier methods that scale cubically with the number of spatial locations (SpatialDE and SPARK) (**Supplementary Figure S20C**). Here, we subsampled the number of spatial locations in the Visium human DLPFC dataset (200, 500, 1000, 2000, and all 3639 spatial locations),

ran each method (nnSVG, SPARK-X, SPARK, and SpatialDE) on 2 genes, and plotted the runtimes (note cubic scale on vertical axis). The figure clearly demonstrates that SpatialDE and SPARK scale cubically (i.e. linear in the cubic axis, with some additional overhead at small numbers of spatial locations for SPARK), while nnSVG and SPARK-X scale sub-cubically. In addition, our earlier computational evaluations (**Figure 3**) demonstrate that nnSVG scales linearly. SPARK-X is the fastest method overall (see also **Supplementary Figure S20D**), but, as demonstrated in the main results, this comes at the cost of substantially lower overall performance due to reduced sensitivity to identify genes with different length scales within the same dataset.

C

New Figure: Supplementary Figure S20C. (Reproduced from manuscript.) (Note cubic scale on vertical axis.)

Response to Reviewer 2

The authors propose an approach “nnSVGs” based on nearest-neighbor Gaussian processes to identify spatially variable genes (SVGs). The proposed approach is superior to the existing methods in that it can identify genes whose expression vary continuously across the entire tissue or within a spatial domain and it allows gene-specific estimates of length scale parameters within the Gaussian process models. Meanwhile, the computational and storage cost of nnSVGs is linear to the number of spatial locations. I find the application of NNGP in identifying SVGs to be interesting and useful. The paper is well-written in general. I have some comments about the technique details of the methodology and data analyses as detailed below.

1. The authors use the exponential covariance function as the kernel of the Gaussian process to model spatial correlation. The hyperparameters (σ^2 , l) in the exponential covariance function are estimated by the maximum likelihood estimates using optimization. My concern here is that when the length scale (parameter l) is very small, a Gaussian process with an exponential kernel can behave like a white noise process. In other words, the estimate of the spatial component of variance σ^2 can be inaccurate when the length scale is very small since the model can hardly distinguish the spatial variance from the white noise variance (nugget). Therefore, I think it will be reasonable to setup a lower bound for the length scale when finding the maximum likelihood estimates. According to Figure 1B, the estimated length scale can be very close to 0. It will be interesting to see whether a lower bound for l can improve the performance of nnSVG or not.

Thank you for this suggestion. Due to the nature of the optimization used for the underlying model fitting (implemented in the BRISC R package), we are unable to adapt the algorithm to set a minimum bound for the length scale parameter estimates. In general, restricting the length scale parameter would imply a composite null hypothesis ($H_0: \sigma^2 = 0$ or $\text{length_scale} < \epsilon$) -- calculating the distribution of the test statistics under this composite null would be difficult, in addition to implementing the constrained optimization within the software. In addition, we note that in our evaluations, we observe some known biologically meaningful SVGs with very small estimated length scale (e.g. *NPY*, which has an estimated length scale of 0.015; see **Figure 1A-B**), so we aim to not impose a strict lower bound at a fixed value such as 0.01.

However, we agree with the reviewer’s point that genes with extremely small length scales (e.g. <0.01) are unlikely to be biologically meaningful, and may be artifacts of the model fitting process. Our approach to reducing the impact of such “noise” genes on the overall results has been to include a filtering step to remove genes with extremely low expression. As demonstrated in **Supplementary Figure S19**, the impact of this filtering procedure is to substantially reduce the proportion of genes with p-values exactly equal to 1 (within rounding).

In order to further investigate the reviewer’s question, we have now performed additional analyses to investigate the results for genes with extremely small length scales (<0.01). **Supplementary Figure S4** compares the rank distributions of the top 3,396 SVGs (equal to the number of genes that pass filtering, when filtering is included), with extremely small length scales (<0.01) and larger length scales, with and without filtering, for the Visium human DLPFC dataset. This shows that, when filtering is included, there are no genes with extremely small length scales (<0.01) within the top 1000 ranked SVGs

(**Supplementary Figure S4A**). By contrast, without filtering, the top 1000 SVGs include several genes with extremely small length scales (<0.01) (**Supplementary Figure S4B**). (In total, we observe 132 and 756 genes with length scales <0.01 , with and without filtering, respectively.) Therefore, since we are largely focused on these top-ranked SVGs, the gene filtering largely addresses the potential issue of low-expressed noise genes with extremely small length scales interfering with the final ranking of top SVGs.

We note that the adaptation to set a minimum bound for length scales could be useful in the context of future work to develop a method to jointly model SVGs across multiple samples (tissue sections), where the estimated parameter estimates may be more robust and less noisy. We have noted this point for future work in the Discussion.

New Figure: Supplementary Figure S4. (Reproduced from manuscript.)

2. The data analyses in Section 2 imply that the proposed nnSVG outperforms SPARK-X but is in general inferior to the baseline method HVGs in recovering SVGs. Table 1 indicates that HVGs is less expensive than nnSVG. Though the authors point out that nnSVG is more flexible than HVGs, it is still not very clear to me why it is important to use nnSVG rather than HVGs. It will be better if the authors can highlight more on why and when one should use nnSVG instead of HVGs (and other methods).

Overall, our results demonstrate that nnSVG gives the best balance of performance across the different datasets that we used for evaluations, while scaling linearly with the number of spatial locations. We agree that HVGs outperforms nnSVG in the Visium human DLPFC dataset (**Figure 1**), and we included this comparison to provide readers with information on the usefulness of methods to identify SVGs compared to simpler baseline methods that either do or do not take spatial information into account (Moran's I and HVGs, respectively). In fact, we note that most previous papers introducing methods to identify SVGs (e.g. SpatialIDE, SPARK, and SPARK-X) did not compare against simpler baseline methods in this level of detail, so this comparison is an important contribution of our paper -- clearly demonstrating the amount of improvement compared to the much simpler strategy of using (non-spatial) HVGs in these datasets.

However, while HVGs (non-spatial baseline method) performs very well in the Visium human DLPFC dataset, **Figure 2** demonstrates that HVGs fails to identify the known SVGs within the spatial domain of interest (CA3 region of hippocampus) in the Slide-seqV2 mouse hippocampus (HPC) dataset. The two known highly informative genes in **Figure 2** are not identified within the top 1000 HVGs, and the alternative spatial baseline method (Moran's I) does not perform well either. By contrast, both nnSVG and SPARK-X perform well in this dataset (**Figure 2**).

In order to further address the reviewer's question and expand on this result, we have now also included evaluations for an extended list of 74 known SVGs from prior analyses of this dataset (including the two genes analyzed previously) (Cable et al. 2021). Specifically, we evaluated how many of these 74 genes were included within the lists of top 1000 SVGs or top 1000 HVGs from each method (nnSVG, SPARK-X, HVGs, and Moran's I) (**Supplementary Figure S12**). This showed that nnSVG recovered the highest number (27 out of 74), followed by SPARK-X and Moran's I (23 out of 74), while HVGs recovered only 3 out of the 74 genes. These results are consistent with the main results in **Figure 2**, showing that nnSVG substantially outperforms HVGs at the task of identifying SVGs within the known spatial domain (CA3 region of hippocampus) within this dataset. As mentioned in the Discussion, the simple (non-spatial) baseline HVGs performs well for identifying SVGs that are related to spatial distributions of cell types, but does not perform well for more subtle patterns of expression such as gradients within spatial domains.

New Figure: Supplementary Figure S12. (Reproduced from manuscript.)

Some notes about writing:

1. The sentence from line 173 to 176 is confusing.

We have re-worded this sentence (and added additional information in the following sentence, based on comments by Reviewer 3).

2. Line 436-447 is exactly the same as line 88-89. A little rewrite might be better.

We have slightly re-worded this paragraph. In addition, we note that we have included some repetition in these sections in order to provide both (i) a short summary of the methods (in the main text) for readers who are mainly interested in application of the method, and (ii) a complete description of the methods (in Methods) for readers interested in the methodology, while (iii) meeting the journal length limits for the main text. While we acknowledge this results in some repetition, in our view, this provides additional clarity for both types of readers.

I hope the authors find these comments helpful for improving their manuscript.

Response to Reviewer 3

Summary:

This manuscript outlines a new method, nnSVG, designed for the detection of spatially variable genes (SVGs) in spatially resolved transcriptomics data. The genes can exhibit spatial variance either across the whole tissue or within pre-annotated regions, depending on the design matrix used during analysis. The method relies on a well-established idea of using Gaussian Processes (GPs) to model gene expression (presented by Svensson et al. in 2018) and the use of a likelihood-ratio test (LR test) -- comparing with a linear model -- to determine whether spatial information is relevant to the expression. While relying on already existing concepts, the authors introduce new and relevant elements in their method; the most notable one being the use of a Nearest Neighbor Gaussian Process (NNGP) to obtain linear (w.r.t. observations) scaling rather than the cubic one observed in standard GP inference. They also use gene-specific estimates of the length scale parameter, something which should better account for differences in expression patterns between genes involved in disparate biological processes.

Comments:

I found this manuscript to be extremely well-written and pleasant to read, everything from structure to language was clear. Section 2.2 “Key Innovations of nnSVG” reminds me of similar sections often found in papers from the machine learning community, this is something I really appreciate and believe helps the reader understand what sets this method apart from already existing ones. My only complaint, with respect to the text, would be that section 2.1 is more or less identical to parts of the Methods section. While I personally like having the more technical aspects of a method as a part of the main text, it might not appeal to the average reader, and it could be wise to compress section 2.1 and refer to the Methods section for more details. Having given the authors some praise, there are also some concerns and comments that I’d like to share, these will be given below.

Thank you for these comments. Regarding Section 2.1 vs. the Methods section, we included both sections for two reasons: (i) to provide both an accessible summary of the methods for readers who are mainly interested in practical applications of the method (Section 2.1) as well as a detailed explanation of methods for methodological readers (Methods), and (ii) to meet the journal length requirements (the detailed Methods section is too long to include in main text). While we acknowledge that this creates some duplication, we have kept this structure so that we can continue to meet both of these objectives.

Major comments:

1. Impact: While I like the technical innovation of this method, I’m less convinced of its impact. To elaborate, in most of the analyses it seems as if nnSVG produces similar results to what the authors refer to as baseline methods, but with a much increased computational cost. Unless I’ve misunderstood parts of the manuscript, I find no compelling evidence that simply using the variance or other simpler metrics to obtain non-spatial HVG genes wouldn’t be enough for a successful downstream analysis.

I realize this comment rather critiques the idea of SVGs than the authors' method, which might seem unfair given how plenty of other methods have been published on the topic. However, as the topic becomes more saturated, I feel as if the need to really show why such methods are necessary also increases. Furthermore, when it comes to the application of finding spatially variable genes within certain domains, one could argue that something similar could be done using the variance, where the observations included in the variance calculation are restricted to those within a certain region of interest. Analyses that the authors could make in order to make a more convincing case would be, but are not limited to:

- a. Exemplifying how there's a qualitative difference in the downstream results when using genes from nnSVG compared to HVGs, preferably to the benefit of the former.
- b. Show that several essential genes, and not only a handful, are lost if only using HVGs

We agree with the reviewer that demonstrating improvement against simpler baseline methods is a crucial part of developing and evaluating a new method. In fact, the detailed comparison against baseline methods (HVGs and Moran's I) represents an important contribution of our paper -- previous papers introducing new methods for SVGs did not compare against simple baseline methods in a similar amount of detail, which makes it difficult for readers to establish whether the more complex methods to identify SVGs are needed at all. We have included these detailed baseline comparisons for transparency and to aid readers in deciding which methods are most suitable for their analyses. We have included an additional sentence in the Discussion to further emphasize this contribution.

To address the reviewer's questions, we have now included additional analyses to demonstrate the improvement when using nnSVG to identify SVGs, compared to the simpler (non-spatial) HVGs baseline. Specifically, we have included the following new analyses:

(a) As described also above for Reviewer 1, we have evaluated clustering performance when using the list of top SVGs from nnSVG as the input for downstream clustering (**Supplementary Figure S15A-D**). We calculated the top 50 principal components (PCs) on either the top 1000 SVGs from each method or the top 1000 HVGs, and then applied standard graph-based clustering from scRNA-seq workflows, for the Visium human DLPFC dataset. We then calculated the adjusted Rand index to compare the similarity between the cluster labels and the manually annotated cortical layer labels available in this dataset (**Supplementary Figure S15E**). This showed that nnSVG and Moran's I (which both take spatial information into account) outperformed HVGs (non-spatial baseline method), as expected. In addition, nnSVG outperformed SPARK-X, which is consistent with the main results showing that the SVGs from nnSVG more closely match the known biological structure in this dataset, compared to SPARK-X (**Figure 1** and **Supplementary Figures S5-S6**). In addition, we note that a recent benchmarking paper has also demonstrated that including SVGs improves downstream clustering performance in spatially-resolved transcriptomics datasets, compared to clustering on HVGs (Li et al. 2022) -- we have added an additional sentence and citation regarding this paper in the Discussion.

(b) We have also extended the analyses for the Slide-seqV2 mouse hippocampus (HPC) dataset to include a longer list of genes (instead of relying only on the two genes shown in **Figure 2**). Specifically, we sourced an extended list of 74 known SVGs from prior analyses of this dataset (including the two previous genes) (Cable et al. 2021), and evaluated how many of these 74 genes were included within the lists of top 1000 SVGs or top 1000 HVGs from each method (nnSVG, SPARK-X, HVGs, and

Moran's I) (**Supplementary Figure S12**). This showed that nnSVG recovered the highest number (27 out of 74), followed by SPARK-X and Moran's I (23 out of 74), while HVGs recovered only 3 out of the 74 genes. These results are consistent with the main results in **Figure 2**, showing that nnSVG substantially outperforms HVGs at the task of identifying SVGs within the known spatial domain (CA3 region of hippocampus) within this dataset.

New Figure: Supplementary Figure S15. (Reproduced from manuscript.)

New Figure: Supplementary Figure S12. (Reproduced from manuscript.)

2. False positives: Another concern of mine is that of false positives. While it seems as if nnSVG is able to identify genes with spatial variance, it feels as if it's fairly promiscuous in some analyses: for example, in the DLPFC data set 60% of the genes are considered as having a statistically significant spatial variance. I understand the difficulty of reporting false positives for this kind of task, as there is no proper ground truth. However, I would maybe suggest the authors generate more complex synthetic data (not just a shuffling of the spots, which breaks all spatial structure). Maybe some synthetic data where the generative model is that of a GP could be used, one idea here would be to induce spatial dependence in only one or a few regions to see how the method handles this (not having any prior annotations of these methods).

We agree the question of false positives is important, and our previous simulations evaluated this only briefly. We have now included an additional extended set of simulations to investigate this question (together with the next section below). We built a simulation framework that simulated a set of SVGs with regions of high expression vs. background expression for simulated genes with varying length scale and expression strength, and then evaluated the true positive rate (TPR) and false positive rate (FPR) for identifying the simulated subset of SVGs (**Supplementary Figure S16**). We obtained empirical parameters for the simulation from the Visium human DLPFC dataset -- mean and variance of log-transformed normalized counts for the known SVG *MOBP* in expressed regions (white matter) and non-expressed regions (cortical layers) respectively, as well as proportions of sparsity within both regions. Then, we simulated 1000 genes, consisting of 100 true SVGs with regions of high expression and low expression, and 900 non-SVGs with background noise only. For the SVGs, we varied the length scale by simulating circular regions of high expression with radius 0.25, 0.125, and 0.025 of the width of the tissue section. We also varied the expression strength as 1, 1/3, and 1/10 times the difference between the regions of high and low expression above the background noise for *MOBP*. **Supplementary Figure S16A** shows the coordinate masks for the simulated regions, and **Supplementary Figure S16B** shows the expression values (logcounts).

Our evaluations for these simulations (**Supplementary Figure S16C**) showed that nnSVG achieved very high TPR, even for the scenarios with relatively low expression strength. Perfect TPR was achieved for all scenarios with large length scale (panels in top row) and medium length scale (middle row). For the small length scale (bottom row), TPR dropped off at the medium expression strength, and eventually reached near zero in the most difficult scenario (small length scale, low expression strength).

Regarding false positives, these simulations demonstrated that nnSVG is conservative overall -- in the "medium length scale, medium expression strength" scenario (middle panel), nnSVG achieves FPR of 0.003, 0.016, and 0.031 at the nominal p-value thresholds of 0.01, 0.05, and 0.1 (**Supplementary Figure S16C**). This result is consistent with our earlier results from the permutation null simulation in **Supplementary Figure S18C**. Overall, we believe this performance is acceptable for the primary purpose of identifying lists of top-ranked SVGs, and we believe these additional simulation results provide important additional context on the statistical performance of our method.

New Figure: Supplementary Figure S16. (Reproduced from manuscript.)

3. Synthetic data: Continuing on the topic of synthetic data, in contrast to the full permutation study described in section 2.5.2. It would be interesting to see whether the method ranks genes in the correct order. For this purpose, I'd suggest an ablation study similar to what is described in: <https://academic.oup.com/bioinformatics/article/37/17/2644/6168120> (see Figure 1).

Thank you for this suggestion. Inspired by the ablation study in the referenced paper, we have built a second set of simulations to address this question (**Supplementary Figure S17**). For this second simulation, we extended the “medium length scale, medium expression strength” scenario from the first

simulation (described above). We designed the ablation setup to shuffle an increasing subset of spatial coordinates in a series of steps representing progressively harder simulation scenarios. Specifically, we simulated 11 scenarios, increasing the proportion of shuffled spatial coordinates by 10% at each step (0%, 10%, 20%, ..., 100%). **Supplementary Figure S17A** shows the shuffled spatial coordinate masks for these scenarios, and **Supplementary Figure S17B** shows the expression values. We then evaluated the TPR at each scenario (**Supplementary Figure S17C**), which showed that nnSVG performed very well even at highly shuffled scenarios (up to 60%). TPR started reducing at 70% shuffled coordinates, and reached near zero by 90% shuffled coordinates. Overall, this demonstrates that nnSVG is highly robust to noise in the spatial patterns of expression, which provides useful additional context for readers.

New Figure: Supplementary Figure S17. (Reproduced from manuscript.)

- Multiple sections: Since SRT methods are becoming ever more popular, there's a trend toward generating larger data sets. Alas, the mouse OB section that the authors use stems from a larger set of 12 sections. My point is that these days, it's rare that we analyze a single section in

isolation. I, therefore, have two questions pertaining to this matter:

- a. How are multiple sections analyzed, is it done one-by-one or jointly? When using non-spatial metrics to identify variable genes, this is a trivial question (as observations are treated as independent no matter if they belong to the same or different tissue sections). I could envision a joint analysis where the entries between observations from non-same sections are nulled, but this also becomes somewhat untrue if these are adjacent sections or if they represent the same region. In a one-by-one analysis, I guess one could take the union (or intersection) of all the SVG genes. Another idea is to transfer the expression to a common coordinate framework (CCF), or in the case of highly similar sections simply align them and create a “consensus slide”, to then apply nnSVG in the shared space. Either way, I would encourage the authors to elaborate on multi-section analysis and how their method handles this.
- b. If the sections in a larger data set are analyzed one by one, how robust are these results? For example, if the authors were to analyze all 12 mouse OB sections or all the sections in the DLPFC data set, how consistent would the results and ranks be? This could, for example, be evaluated with Kendall’s Tau. Actually, this question of robustness is something that I’d like the authors to address no matter the answer to a).

We agree that spatially-resolved transcriptomics datasets consisting of multiple sections (replicates as well as samples from multiple biological conditions) will become an increasing focus for future work, both experimental and methodological. In the current implementation of our method, we have focused on a single tissue section at a time. However, in ongoing subsequent work, we are actively developing a principled approach for analyzing data from multiple sections at a time. We have included additional wording in the Discussion to highlight this direction for future work by ourselves and others.

(a) In addition, we have developed an initial approach to apply the current version of nnSVG to data from multiple sections. Specifically, we first apply nnSVG individually to each section to generate a list of top SVGs per section. Then, we calculate the average ranks per gene across all sections, and re-calculate the ranks of these averaged ranks to use as the final combined ranking of SVGs across sections. In addition, we calculate the number of times each gene ranks within the top 100 SVGs across sections, in order to exclude (or further investigate) any genes that appear as highly ranked only within one or a small number of sections. We successfully applied this approach in a recent collaboration to identify SVGs within the locus coeruleus region in human brain samples (Weber and Divecha et al. 2022). We have also included a detailed example demonstrating this approach in the latest development version of the nnSVG package documentation (vignette), available from Bioconductor (development version) and GitHub (https://bioconductor.org/packages/devel/bioc/vignettes/nnSVG/inst/doc/nnSVG.html#43_Multiple_samples).

(b) To address the reviewer’s question, we have also performed additional analyses in the Visium human DLPFC dataset, applying nnSVG individually to each tissue section and comparing the consistency of the rankings per section (**Supplementary Figure S23**). We calculated the Spearman correlation between the rankings for each pair of samples (12 samples total) in this dataset, and found that these correlations were relatively high (>0.8) between samples *within* donors 1 and 3 respectively, moderate (>0.75) between samples *between* donors 1 and 3, and lower within donor 2 and between

donor 2 and the other donors (**Supplementary Figure 23A**). This reflects the known biological structure from previous analyses of this dataset, which showed that the samples from donor 1 and 3 had all cortical layers present, while the samples from donor 2 were missing several cortical layers (**Supplementary Figure 23B**). We also visualized the comparison of ranks for the samples with the highest and lowest correlation with sample 151673 (the sample used in the main results in **Figure 1**) respectively **Supplementary Figure 23C-D**) to further demonstrate these results.

New Figure: Supplementary Figure S23. (Reproduced from manuscript.)

5. Run-times: I would also like to see run-time estimates when using a more complex design matrix than simply an intercept. I expect this to increase the run-time somewhat as more parameters need to be fitted.

We have included additional analyses evaluating the runtime when including a design matrix, using the Slide-seqV2 mouse HPC dataset (**Supplementary Figure S20A-B**). Here, we ran nnSVG with and without the design matrix for known spatial domains, using a single gene and subsetting the number of

spatial locations, and running nnSVG 10 times at each number of spatial coordinates to evaluate random variation. We observe only slightly higher runtimes with the covariates (**Supplementary Figure 20A**), compared to without covariates (**Supplementary Figure 20B**).

New Figure: Supplementary Figure S23A-B. (Reproduced from manuscript.)

6. Run-times: Initially, upon reading the Introduction and Abstract, I thought that this manuscript would present some stellar run-times, as the authors obviously have put a lot of thought into this topic. However, I was somewhat negatively surprised to see the -- apologies for the expression, but it's the only true assessment -- poor run-time results. The nnSVG method takes on average 133x longer time to analyze the data compared to SPARK-X. HVG and SVG identification is usually given little room in SRT studies, and the main purpose of this task is usually to select genes for downstream analysis; I find it somewhat hard to justify a 5h wait just to get a list of spatially variable genes, this while also requiring 10 cores (which many personal computers do not even have). This makes me disagree with Table 1, where nnSVG is reported to have "intermediate runtimes", if only comparing with the 4 listed methods, I would definitely put this value as "not"; however if an older method like SpatialDE was included in the comparison, I could agree with the current "intermediate" classification. Could the authors either:
 - a. Motivate how this run-time is acceptable, especially in the context of what is gained from having these genes compared to standard HVG genes (relating to my question above)
 - b. Improve the run-time, I'm not familiar with all of the assumptions in the NNGP model, but could you perhaps use inducing points? Looking at the referenced paper, it seems as if Gibbs sampling is used (though I'm not sure about the implementation), maybe this could be replaced by a more efficient algorithm that could be sped up by GPU acceleration?

We agree that the runtimes for nnSVG are intermediate overall, and that this may not have been clear from the summary table (which is missing the context that the earlier methods scale cubically with the number of spatial locations). Crucially, the runtimes for nnSVG scale linearly with the number of spatial locations, so the method can be applied to large datasets with manageable runtime (as we have demonstrated in **Figure 3**). In addition, we note that in the comparison with SPARK-X, while SPARK-X

is extremely fast, this comes at the expense of drastically lower performance -- as demonstrated in **Figure 1**, SPARK-X completely misses the known SVGs in this dataset with small length scales. In our view, SPARK-X gives up too much performance in order to achieve fast runtimes -- while nnSVG achieves the balance between linear scalability and excellent performance for identifying nnSVG across a range of datasets. Similarly, HVGs does not adequately identify the known SVGs in the Slide-seqV2 mouse HPC dataset (**Figure 2** and **Supplementary Figure S12**, as described above).

In order to provide further context for readers and to make this tradeoff more clear, we have: (i) included additional results comparing runtimes across methods and demonstrating the cubic scaling for the earlier methods (SpatialIDE and the original version of SPARK) (**Supplementary Figure 20C**) (see also above for Reviewer 1), and (ii) updated the summary table to include additional rows for the earlier cubically scaling methods (SpatialIDE and SPARK).

With regard to inducing points -- these are a class of low-rank approximations commonly used to speed up GPs, commonly referred to as predictive processes in the spatial statistics literature (Banerjee et al. 2008). Previous work has shown that using inducing points performs far worse than NNGPs given similar computing time (due to over-smoothing), while increasing the number of inducing points to improve performance increases the computational cost cubically in the number of inducing points (Datta et al. 2016, Heaton et al. 2020), so here we preferred to apply the NNGPs approach.

We have also included wording in the Discussion noting that further runtime optimizations would be an excellent avenue for future work, either by ourselves or others.

New Figure: Supplementary Figure S20C. (Reproduced from manuscript.) (Note cubic scale on vertical axis.)

Minor comments

1. In their abstract, the authors write “to identify spatially variable genes is a key step during analyses of spatially resolved transcriptomics data”. While this might be true in the future, I’m

keen to argue that such is not the case at the present time. Most SRT-based studies settle with just HVGs and do not seem to rely on SVGs.

We have clarified this language by including the words “spatially variable genes or other biologically informative genes”, i.e. referring to the general case of either SVGs or HVGs.

2. On line 21 the authors write “up to thousands of genes at near- or sub-cellular resolution”. To me “up to thousands” does not really convey the fact that the full transcriptome (or at least polyadenylated transcriptome) can be captured in many methods, maybe reconsider this choice of wording.

We have clarified this wording by adding the words “full transcriptome”, i.e. to refer to the general case of either full-transcriptome methods (e.g. sequencing-based methods such as Visium) as well as targeted methods with up to thousands of targeted genes (e.g. MERSCOPE and Xenium).

3. In most existing methods for detection of SVGs there seems to be a preference for ranking by adjusted p-value, is there any particular reason that the authors chose to rank their genes by the LR statistic?

We note that while ranking by the p-value and likelihood ratio (LR) statistic are equivalent in theory, in our view, there are several advantages in practice to ranking by the LR statistic: (i) for some genes with extremely strong spatial patterns, we observe p-values exactly equal to 0 due to rounding and machine precision together with our use of an approximate chi-square test -- these genes cannot be distinguished by p-value but can still be uniquely ranked by their LR statistics; (ii) depending on the stringency of filtering for low-expressed genes, we also observe some proportion of genes with p-values exactly equal to 1 (**Supplementary Figure S19**), which similarly cannot be distinguished by p-value but can be uniquely ranked by their LR statistics; and (iii) overall, we prefer to focus on the ranking (e.g. top 100 SVGs for further individual investigation, or top 1000 SVGs used as input for downstream clustering) instead of focusing on the set of significant SVGs -- using the LR statistic for the final ranking lets us more easily emphasize this focus in the final results. Note that for the majority of the genes (except those with p-values exactly equal to 0 or 1), the rankings by LR statistic and p-value are one-to-one, so the results for these genes are identical.

4. In section 2.3.1 it's reported that out 3396 genes included in the analysis, 2049 were considered as SVG, and subsequently, 133 of the 134 known cortical layer-associated SVGs were correctly identified. Maybe the authors could provide a p-value for how likely this is to happen by chance; that is if I select 2049 genes from 3396 ones, how likely am I to get 133 “correct ones” out of 134. Fisher's exact test would probably be befitting for this task.

Using Fisher's exact test, the p-value for this result by chance (assuming independently selected genes) is $p < 10^{-16}$. We have included this value in the text.

5. On lines 212-213 it's stated that the OB mouse data set contains a smaller set of spatial locations; since not everyone is familiar with the old ST platform, it might be beneficial to also mention that, in addition to this, the resolution is lower than the more familiar Visium platform.

We have added this clarification in the text.

6. On lines 260-261, concerning the Slide-seqV2 analysis, it's stated that HVG baseline methods do not perform as well as in the Visium data set. Is this statement solely based on whether the two genes were included in the set of the top 1000 most highly ranked ones? If that is true, I think that's a bit of a drastic conclusion, and I feel as if the ability to correctly rank genes should be assessed using a larger gene set.

We agree this is a fair point, and have significantly expanded the results for this dataset by including an additional extended list of known SVGs (74 genes) from previous analyses (Cable et al. 2021), as described above under major comments (**Supplementary Figure S12**). The inclusion of this extended gene list has significantly strengthened the evidence for our claim that nnSVG outperforms the HVGs baseline method for identifying SVGs within the known spatial domain in this dataset.

7. It would be great if the authors reported the improved scaling properties when exchanging the standard GP for an NNGP. If I read the material correctly, it shifts from $O(n^3)$ to $O((n+k)m^3)$ where n = number of observations, k = number of reference points, m = number of neighbors.

The computational complexity of the model fitting per iteration is $O(n * m^3)$, where n = number of spatial locations, m = number of nearest neighbors (default = 10), and the initial steps of ordering coordinates and calculating nearest neighbors are performed once only and are re-used for all genes. We have included a sentence with this information in the Methods.

Additional Updates

We have also made the following additional updates to the analyses and manuscript:

- The preprocessing steps in the analyses have been updated to use library size normalization for all datasets. In the previous version of the manuscript, we used an alternative normalization methodology (normalization by deconvolution, which is recommended for scRNA-seq data; Amezquita et al. 2019) for the human DLPFC dataset. However, normalization by deconvolution requires an initial pre-clustering step, which may be biased in spatial data due to the presence of multiple cell types per spot. Therefore, we prefer the simpler library size normalization methodology, which does not require pre-clustering. This change is also consistent with other recent work (e.g. Weber and Divecha et al. 2022). This change resulted in some minor changes to the rankings and correlations for individual SVGs for the human DLPFC dataset. There were no changes to the overall performance rankings of the methods. The nnSVG package documentation and tutorial have also been updated to reflect this change.
- Several minor typos, clarifications, and additional references in the text (highlighted with blue font in the reviewer copy of the manuscript).

References

Amezquita et al. (2019), *Orchestrating single-cell analysis with Bioconductor*, Nature Methods, 17, 137-145.

Banerjee et al. (2008), *Gaussian predictive process models for large spatial data sets*, Journal of the Royal Statistical Society: Series B (Statistical Methodology), 70(4), 825-848.

Cable et al. (2021), *Robust decomposition of cell type mixtures in spatial transcriptomics*, Nature Biotechnology, 1(1).

Datta et al. (2016), *Hierarchical nearest-neighbor Gaussian process models for large geostatistical datasets*, Journal of the American Statistical Association, 111(514), 800-812.

Heaton et al. (2019), *A case study competition among methods for analyzing large spatial data*, Journal of Agricultural, Biological and Environmental Statistics, 24, 398-425.

Li et al. (2022), *Benchmarking computational integration methods for spatial transcriptomics data*, bioRxiv (preprint).

Weber and Divecha et al. (2022), *The gene expression landscape of the human locus coeruleus revealed by single-nucleus and spatially-resolved transcriptomics*, Reviewed Preprint (eLife).

REVIEWERS' COMMENTS

Reviewer #1 (Remarks to the Author):

All of my comments have been addressed. I congratulate the authors for their great work.

Reviewer #2 (Remarks to the Author):

I have carefully reviewed the revised manuscript and I am pleased to say that the authors have addressed all my previous concerns. Furthermore, I find the additional updates strengthen the overall contributions of the paper. In light of these improvements, I believe that the manuscript now meets the standards for publication in Nature Communications.

Reviewer #4 (Remarks to the Author):

The authors have done a great job in revising their manuscript and in doing so they have addressed all the remaining concerns raised by the previous reviewers.

Reviewer 1 Comments

Reviewer #1 (Remarks to the Author):

All of my comments have been addressed. I congratulate the authors for their great work.

Reviewer 2 Comments

Reviewer #2 (Remarks to the Author):

I have carefully reviewed the revised manuscript and I am pleased to say that the authors have addressed all my previous concerns. Furthermore, I find the additional updates strengthen the overall contributions of the paper. In light of these improvements, I believe that the manuscript now meets the standards for publication in Nature Communications.

Reviewer 4 Comments

Reviewer #4 (Remarks to the Author):

The authors have done a great job in revising their manuscript and in doing so they have addressed all the remaining concerns raised by the previous reviewers.